# LiTo: Surface Light Field Tokenization

**Jen-Hao Rick Chang**[*]    **Xiaoming Zhao**[*]    **Dorian Chan**    **Oncel Tuzel**
Apple

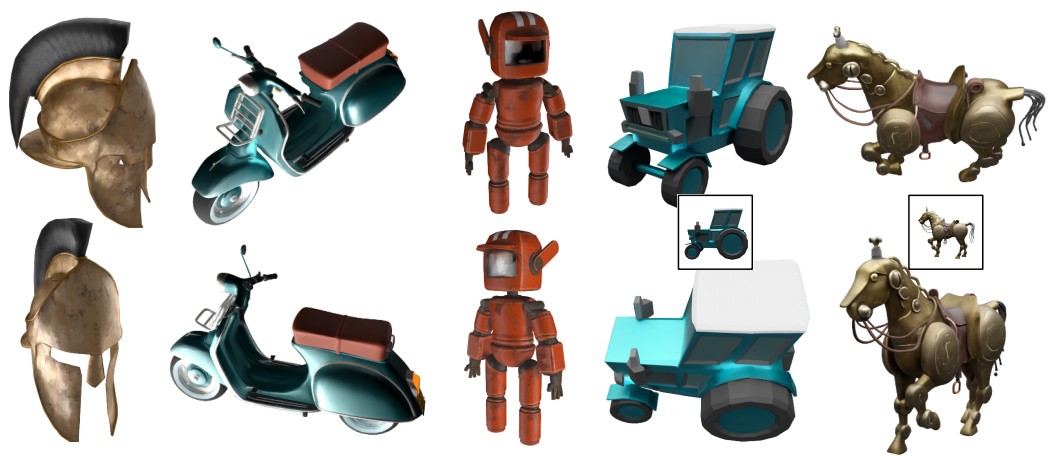

Figure 1: LiTo tokenizes surface light fields into a latent representation. It models 3D geometry and view-dependent appearance such as specular reflection. The figure shows reconstructions (first 3 columns) and single-image-to-3D results (last two columns). Mesh credit: Anthony Schmidt (2016); @sanyabeast (2021); LLOYDO (2019); brysew (2015); Osho (2018). See more on the project page.

## Abstract

We propose a 3D latent representation that jointly models object geometry and view-dependent appearance. Most prior works focus on either reconstructing 3D geometry or predicting view-independent diffuse appearance, and thus struggle to capture realistic view-dependent effects. Our approach leverages that RGB-depth images provide samples of a surface light field. By encoding random subsamples of this surface light field into a compact set of latent vectors, our model learns to represent both geometry and appearance within a unified 3D latent space. This representation reproduces view-dependent effects such as specular highlights and Fresnel reflections under complex lighting. We further train a latent flow matching model on this representation to learn its distribution conditioned on a single input image, enabling the generation of 3D objects with appearances consistent with the lighting and materials in the input. Experiments show that our approach achieves higher visual quality and better input fidelity than existing methods.

## 1    Introduction

The world is filled with objects that vary widely in shape and material. Some are smooth and reflective, while others are rough, detailed or even translucent. Even familiar objects can appear differently from different viewpoints as light creates reflections and subtle color changes across their surfaces. Capturing this richness is important for building generative models of realistic objects. To do so, we need representations that can model both the underlying 3D geometry of real-world objects as well as their view-dependent appearance.

However, today in machine learning, most existing 3D representations tackle only part of this problem. Many methods are designed to capture geometry alone (He et al., 2025; Li et al., 2025a; Chang

---

[*]Indicates equal contribution. See Sec. G for a detailed breakdown of individual contributions.

et al., 2024), aiming to recover the overall shape of objects. Other approaches (Xiang et al., 2025) include appearance information, but treat it as view-independent diffuse color. As a result, these models struggle to represent view-dependent effects such as reflections, highlights, or subtle changes in shading that are important for realistic appearance.

In this work, we aim to model both the 3D geometry and the view-dependent appearances of objects. We introduce a 3D latent representation that encodes a *surface light field* into a compact set of latent vectors. In summary, rather than encoding geometry and color only, *e.g.* with an input RGB point cloud, we additionally input viewing direction along with surface points and color, to capture how realistic materials change appearance with angle. Because a full surface light field contains highly dense information, we instead provide a random subsample of the surface light field—captured from RGB-depth multiview images—and rely on an encoder to interpolate the missing samples. This approach allows the model to reproduce view-dependent effects such as highlights and Fresnel reflections, that can be visualized via a decoder that outputs Gaussian splats with higher-order spherical harmonics (Kerbl et al., 2023). We evaluate our method by comparing its reconstruction quality against the state-of-the-art 3D latent representations (Xiang et al., 2025; Li et al., 2025a; He et al., 2025; Chen et al., 2025b; Chang et al., 2024), and find that modeling these view-dependent effects improve visual quality without significant degradation in geometric accuracy.

Building on the proposed representation, we train a latent flow matching model that learns the distribution of our 3D latent representations conditioned on a single input image. The generative model learns to infer both geometry and view-dependent appearance from images under different lighting conditions. Given an input image, the model generates a full 3D object whose shape matches the object in the image from the input viewpoint and whose appearance reflects the lighting and view-dependent material properties present in the input. Our approach connects 2D observations to 3D object generation, enabling controllable synthesis of realistic, view-dependent materials from diverse image inputs.

Our work makes the following contributions.

- We introduce a 3D latent representation that captures both geometry and view-dependent appearances by encoding surface light field information into a compact set of latent vectors.

- We design a training framework that jointly supervises geometry and appearance using random subsamples of surface light field data from RGB-depth multiview images, enabling the model to reproduce view-dependent effects such as highlights and fresnel reflections via Gaussian splats with higher-order spherical harmonics.

- We develop a latent flow matching model that learns the distribution of these latent representations conditioned on images, allowing the generation of full 3D objects whose appearances reflect the lighting and materials in the input.

Together, these components enable more accurate reconstruction and better separation of geometry and appearance than existing methods.

## 2    RELATED WORKS

A growing number of recent approaches have explored learning latent 3D representations. In Tab. S1, we summarize and compare their properties, including geometry and appearance modeling mechanisms, data requirements, latent dimensionality, encoder inputs, and training sets. For clarity, we review geometry-only approaches and those that jointly model geometry and appearance separately.

**Geometry-only latent.**    A large body of work focuses on latent representations that model geometry alone. These approaches differ primarily in the underlying 3D signal they encode. PointFlow (Yang et al., 2019), ShapeGF (Cai et al., 2020), and ShapeToken (Chang et al., 2024) learn to model 3D surfaces as 3D distributions. 3DShape2VecSet (Zhang et al., 2023), CLAY (Zhang et al., 2024), TripoSG (Li et al., 2025a), and Hunyuan3D (Zhao et al., 2025), instead model shapes as occupancy or signed distance functions (SDF). Direct3D (Wu et al., 2024), XCube (Ren et al., 2024), LT3SD (Meng et al., 2025), and Make-A-Shape (Hui et al., 2024) embed geometry into dense or sparse voxel grids containing occupancy or SDF values at vertices. While grid-based methods offer structured latents, they face inherent trade-offs between spatial resolution and memory efficiency. A common limitation

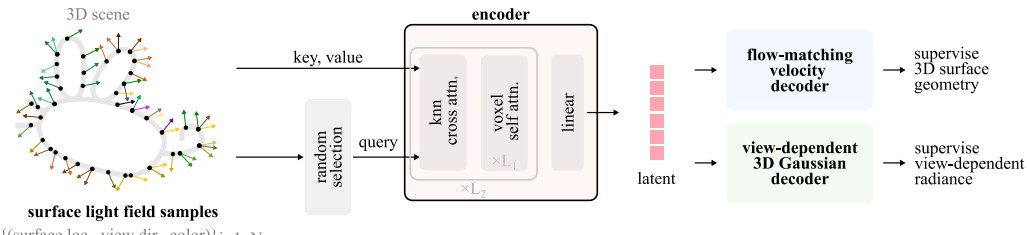

Figure 2: **Overview of the 3D latent representation.** Given samples of the surface light field of the scene, we learn a latent representation that reconstruct the *full* surface light field information. The encoder (pink block) condenses input information into the latent representation. We jointly supervise the latent representation to contain full 3D geometry and view-dependent radiance information beyond the input samples. In the architectures, we design localized attention pattern to improve efficiency and support 1 million input tokens.

when relying on occupancy or SDF, however, is the reliance on significant preprocessing of the training data. Many methods require watertight meshes (Zhang et al., 2023; 2022; 2024), expensive mesh-to-field conversions, or optimization-based radiance-field fitting in order to define consistent supervision signals. Moreover, these methods capture only geometry, without appearance, texture, or view-dependent effects.

**Geometry and appearance latent.** More recently, a smaller set of works has begun to extend latent 3D representations beyond pure geometry to also encode appearance. Two of the most relevant are 3DTopia-XL (Chen et al., 2025b) and TRELLIS (Xiang et al., 2025).

3DTopia-XL introduces the PrimX representation, where each primitive encodes not only geometry through signed distance but also material properties such as RGB color, roughness, and metallicity. This design allows the model to generate textured 3D assets that are ready for physically based rendering. However, PrimX requires an optimization step to construct the primitive representation from meshes before training, making data preparation more demanding.

TRELLIS introduces a Structured LATent (SLAT) representation: a sparse voxel grid fused with dense multiview visual features extracted by a foundation vision model (DINOv2) to provide both geometry and appearance cues. Given the coarse geometry of an object, SLAT is constructed by averaging projected DINOv2 features from all input views. The model decodes SLAT into multiple output 3D formats, including 3D Gaussians, meshes, and radiance fields. To handle the sparsity of SLAT efficiently, TRELLIS employs transformers with windowed attention and sparse 3D convolution, and it is trained at scale on roughly 500K assets from Objaverse-XL and related datasets.

TRELLIS has several limitations relative to our approach. First, SLAT requires coarse occupancy information to be known in advance, so generation is performed in two stages, whereas our latent directly encodes complete object information and supports single-stage generation. Second, TRELLIS encodes only view-independent appearance: multiview features are mean-pooled, discarding angular variation and preventing modeling of view-dependent effects. Finally, TRELLIS generates objects in a canonical coordinate system (*i.e.*, their dataset orientation), which necessitates post-processing to align them with input images. This restriction arises from its reliance on preconstructed axis-aligned voxel grids, which makes coordinate transformations like rotation during training difficult. In contrast, our model takes points as input, which allows us to apply coordinate transformations during training, ensuring generated objects are consistently oriented with respect to the input view (see Fig. 5 and 6).

## 3  METHOD

### 3.1  PRELIMINARY AND NOTATION

The surface light field jointly models both the 3D surfaces of a scene as well as the outgoing radiance from each point on the surface toward every viewing direction. In theory, if the surface light field is perfectly represented, any image captured by a camera at any arbitrary location and

orientation can be directly reconstructed (Wood et al., 2000). We represent the surface light field as a 5D function $\ell(\mathbf{x}, \hat{\mathbf{d}}) : \mathbb{R}^3 \times \mathbb{S}^2 \to \mathbb{R}^3$, where $\mathbf{x} \in \partial\Omega$ is any 3D location on surfaces $\partial\Omega$, $\hat{\mathbf{d}} \in S^2 = \{\mathbf{v} | \mathbf{v} \in \mathbb{R}^3, \|\mathbf{v}\| = 1\}$ is the viewing direction, and $\mathbf{c} \in \mathbb{R}^3$ is the color of the outgoing radiance from $\mathbf{x}$ toward $\hat{\mathbf{d}}$.

We use bold lowercase symbols (*e.g.*, $\mathbf{v}$) to denote vectors, bold lowercase symbols with hats (*e.g.*, $\hat{\mathbf{v}}$) for unit-norm directions, capital letters (*e.g.*, $A$) for matrices or transformations, and calligraphic symbols (*e.g.*, $\mathcal{S}$) for sets.

### 3.2 TOKENIZER OVERVIEW

Our goal is to learn a 3D latent representation that models the surface light field of an object-centric scene with a compact set $\mathcal{S} \triangleq \{\mathbf{s}_j\}_{j=1}^k$, where $\mathbf{s}_j \in \mathbb{R}^d$ is a $d$-dimension latent vector. Fig. 2 shows an overview of our latent representation. Our encoder outputs $\mathcal{S}$ after taking $N$ samples of the surface light field defined in the following as input:

$$\mathcal{X} = \{(\mathbf{x}_i, \hat{\mathbf{d}}_i, \mathbf{c}_i = \ell(\mathbf{x}_i, \hat{\mathbf{d}}_i))\}_{i=1}^N., \tag{1}$$

where $\mathbf{x}_i$, $\hat{\mathbf{d}}_i$, $\mathbf{c}_i$, and $\ell(\cdot, \cdot)$ are defined in Sec. 3.1.

To learn a meaningful representation of the surface light field, we must supervise both the decoded 3D geometry as well as view-dependent radiance. A trivial solution would utilize an autoencoder formulation that directly reconstructs the input $\mathcal{X}$. However, in practice we only have sparse, discrete samples of the surface light field (*e.g.*, as rendered from multiview images of a training object), and thus such an approach may not meaningfully represent the entire continuous function $\ell$. Thus, rather than directly supervising with the surface light field, we instead opt for indirect supervision with carefully-designed loss functions on decoded geometry and view-dependent appearance (as well as the regularization in Sec. E.4):

**Geometry supervision.** We utilize prior work (Chang et al., 2024), which models 3D surfaces as a 3D probabilistic density function that is aligned with the actual surfaces via flow matching. This formulation enables us to model 3D surfaces beyond the input 3D locations. Specifically, the latent $\mathcal{S}$ is trained to parameterize a 3D distribution $p(\mathbf{x}|\mathcal{S})$ that approximates a dirac delta function lying on 3D surfaces in the scene, *i.e.*, $p(\mathbf{x}|\mathcal{S}) \approx \delta(\mathbf{x} \in \partial\Omega)$. The flow matching formulation also optionally allows us to sample $p(\mathbf{x}|\mathcal{S})$ and get a point cloud lying on surfaces during inference, and zero-shot estimate surface normals. The loss function follows that used by Chang et al. (2024):

$$\mathcal{L}_{\text{geo}}(\boldsymbol{\theta}) = \mathbb{E}_{t \sim U(0,1)} \mathbb{E}_{\mathbf{x}} \|V(\mathbf{x}_t; t) - (\mathbf{x} - \boldsymbol{\epsilon})\|^2 \, \mathrm{d}t \, , \tag{2}$$

where $\boldsymbol{\theta}$ is all parameters in the encoder and the decoder, $t$ is the flow-matching time, $U(0, 1)$ is the uniform distribution between 0 and 1, $\boldsymbol{\epsilon}$ is noise sampled from standard normal distribution, $V_\theta(\mathbf{x}_t; t)$ is the flow-matching decoder that estimates the velocity at $\mathbf{x}_t = t \cdot \mathbf{x} + (1 - t) \cdot \boldsymbol{\epsilon}$, and $\mathbf{x}$ is sampled from the surface light field.

**View-dependent radiance supervision.** The supervision of the view-dependent radiance is through rendering multi-view images. Specifically, we convert the latent $\mathcal{S}$ into a set of 3D Gaussians, which models view-dependent color by spherical harmonics, and we render the 3D Gaussians from random viewpoints and compare with ground-truth images. The loss is

$$\mathcal{L}_{\text{radiance}}(\boldsymbol{\theta}) = \mathbb{E}_{H,E} \|I_{\text{est}} - I_{\text{gt}}\|^2 + \lambda \, \text{lpips}\,(I_{\text{est}}, I_{\text{gt}}) \, , \tag{3}$$

where $I_{\text{est}} = \texttt{Render}(D(\mathcal{S}, \mathcal{O}), H, E)$ is the rendered image from 3D Gaussians at camera pose $H$ and intrinsic $E$, $I_{\text{gt}} = \texttt{Render}(\text{object}, H, E)$ is the ground-truth image, $D$ is the Gaussian decoder that will be detailed below, $D(\mathcal{S}, \mathcal{O})$ are the estimated 3D Gaussians given the latent $\mathcal{S}$ and a low-resolution sparse occupancy grid $\mathcal{O}$ constructed from the sampled point cloud or an occupancy estimator, and $\boldsymbol{\theta}$ is all parameters in the encoder and the decoder. In all experiments, we use $\lambda = 0.2$.

In the rest of this section, we discuss the architectures for our surface light-field encoder, geometry decoder and Gaussian decoder in more detail.

### 3.3 ENCODER

We first describe how we sample surface light field to obtain the input to the encoder and the samples for the loss in Eq. (2). Then we detail our encoder architecture.

**Input.** To sample from the surface light field $\ell(\mathbf{x}, \hat{\mathbf{d}})$ in Eq. (1), we need to sample random surface locations and view directions. We achieve this by densely rendering multi-view RGBD images. Since we focus on object-centric scenes, the cameras are placed uniformly on a sphere surrounding the object. The surface location $\mathbf{x}$ can be obtained by back-projecting the depth map, view direction $\hat{\mathbf{d}}_i$ is derived from the pinhole camera model, and $\mathbf{c}_i$ from the pixel color[1]. This operation densely samples both the surfaces and viewing directions and returns $\mathcal{X} = \{(\mathbf{x}_i, \hat{\mathbf{d}}_i, \mathbf{c}_i)\}_{i=1}^N$ in Eq. (1).

In our experiments, we box-normalize the scene to $[-1, 1]$, and we render 150 images of resolution $1036 \times 1036$ with 40 degree field of view, uniformly on a sphere of radius 3.5. This provides 160 million samples of light field $\ell$ introduced in Sec. 3.1, of which we randomly sample $N = 2^{20}$ as our input to the encoder and the rest to serve as the ground-truth to supervise Eq. (2).

**Architecture.** We use Perceiver IO (Jaegle et al., 2022) as our encoder, which is widely used in prior latent 3D representations (Zhang et al., 2023; Chang et al., 2024; Li et al., 2025a). The encoder contains cross and self attention blocks, and the number of initial queries of the first cross attention block determines the number of output latent tokens, *i.e.*, $k = 8192$ in our case discussed in Sec. 3.2. The output of the Perceiver IO is passed to a linear layer to reduce the latent dimension to $d = 32$. Our latent $\mathcal{S}$ is thus a set of $k$ tokens of $d$ dimension (see Sec. 3.2).

To capture enough information from light field $\ell$ introduced in Sec. 3.1, we use $N = 2^{20}$ ($\sim$1 million) samples as input. However, the large number of input makes the typical cross attention in Perceiver IO computationally expensive. We are inspired by the non-overlapping patchification in Vision Transformers (Dosovitskiy et al., 2021), which converts dense pixels into coarse tokens. Instead of using a convolution layer to aggregate information from individual $16 \times 16$ patches into tokens, we use cross attention. However, our inputs are scattered points on 3D surfaces instead of pixels on a regular grid, and it is non-trivial to patchify 3D surfaces.

We design an approximation of 2D patchification on 3D surfaces with K-nearest neighbor. Specifically, given the input samples $\mathcal{X}$ in Eq. (1), we first randomly select $k$ samples as the query $\mathcal{Q}$ to the first cross attention layer, similar to Zhang et al. (2023). The number of samples is equal to the number of latent tokens $k$. To patchify 3D surface, for each sample $\mathbf{x} \in \mathcal{X}$ we find its closest point in $\mathcal{Q}$ in terms of $\ell_2$ distance of $\mathbf{x}$ and assign the index of the closest point to the sample. Finally, during the cross attention, a query only attends to input samples that have its index. This operation can be implemented by standard libraries like `xformers` (Lefaudeux et al., 2022) or `FlashAttention` (Dao, 2024).

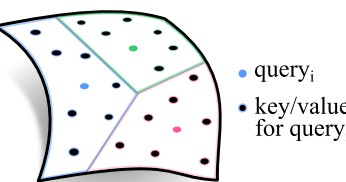

Figure 3: 3D patchification

An illustration is shown in Fig. 3. Note that this is an approximation because we use $\ell_2$ distance of $\mathbf{x}$ instead of geodesic distance. Thus, when there are more than one surface lie in the neighborhood, the query will attend across surfaces. As $\ell_2$ distance is much faster to compute than the geodesic distance, we think it is a good trade-off.

For self attention, we use a voxel-based attention mechanism. Specifically, tokens that lie within the same voxel in a predefined coarse grid attend to each other, and the coarse voxel grid shifts by a half cell width every layer. Unlike TRELLIS (Xiang et al., 2025), whose tokens lie on a voxel grid, our tokens have continuous coordinates and are not grid-aligned. We use a voxel grid only to organize self-attention. Overall, the encoder has 59.2 million parameters (see Fig. S2). Together with decoders below, the model is trained with 256 batch size for 90k iterations on 64 GPUs for 9 days.

## 3.4 DECODER

**Flow-matching velocity decoder.** We utilize the same flow-matching velocity decoder used by Chang et al. (2024). Specifically, it takes the latent $\mathcal{S}$, a 3D location, and flow-matching time as input, and it predicts the flow-matching velocity at the 3D location. To ensure we model a 3D distribution, *i.e.*, $p(\mathbf{x}|\mathcal{S})$, the decoder processes each 3D point independently (only cross attention and point-wise operations are used). The decoder has 8.8 million parameters. See details in Fig. S3.

---

[1]We assume the depth map measures the distance to the first intersection point of the scene, regardless of transparency. For example, in blender, this can be achieved by setting the alpha threshold to be 0.

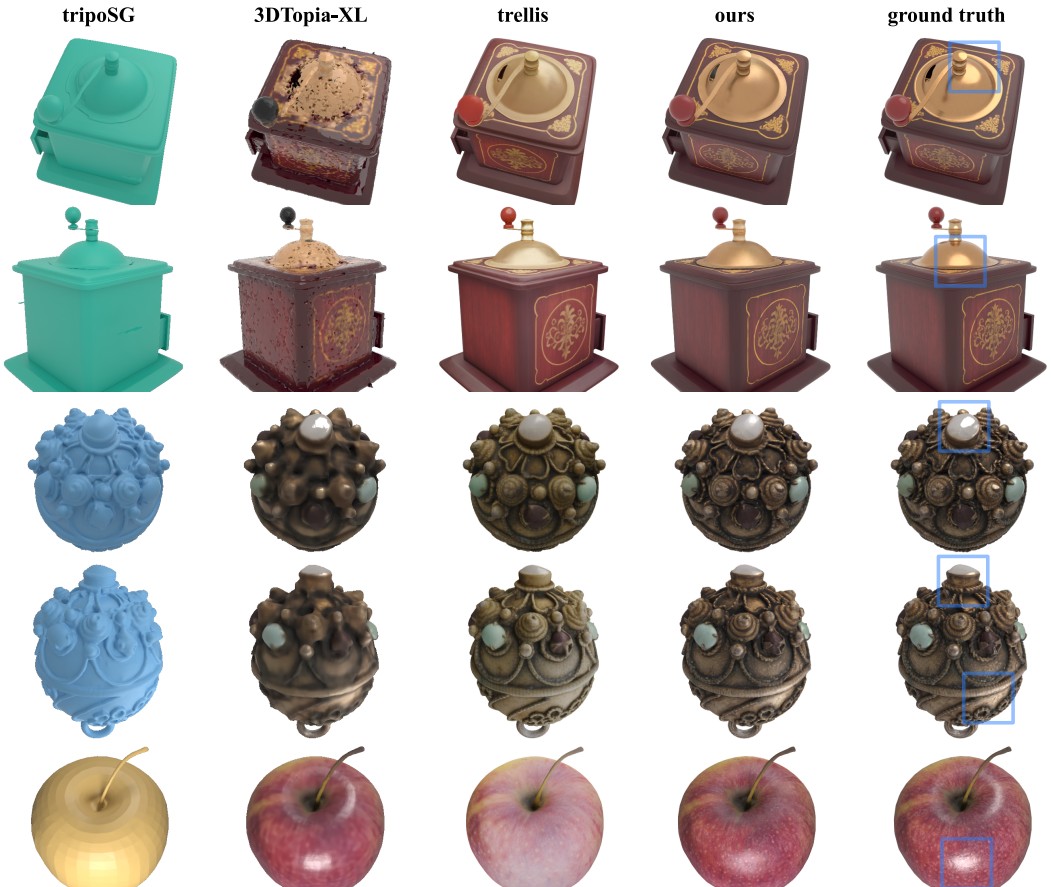

Figure 4: **Reconstruction results on various lighting conditions.** Boxes on ground-truth highlight specular and Fresnel reflection. Please refer to Tab. 1 for quantitative results. Mesh credit: Digital-Souls (2019); 3Dji (2025); Virtual Museums of Małopolska (2020).

**View-dependent Gaussian decoder.** Similar to our encoder, we use a Perceiver IO architecture (Jaegle et al., 2022) for our Gaussian decoder. We use a low-resolution sparse occupancy grid for our initial queries, and cross attend to the predicted latent $\mathcal{S}$. We use a small MLP to output 64 3D Gaussians for each occupied voxels (see Sec. E.3). Unlike past work that only uses Gaussians with view-independent color (Xiang et al., 2025), our decoder predicts Gaussians of spherical harmonics degree 3 for view-dependent radiance. We observe that different harmonic degrees encode distinct appearance characteristics (see Sec. F.1). The decoder has 77.3 million parameters (see Fig. S4).

At training time, we use ground-truth occupancy for the decoder queries, like recent work leveraging structured latent representations (Xiang et al., 2025; He et al., 2025; Wu et al., 2025). After learning the representation, we can either use points sampled from the aforementioned flow-matching geometry decoder or alternatively train a downstream occupancy decoder (see Sec. E.5 and Fig. S5), to directly predict sparse occupied voxels from the encoded latent. Thus, at generation time, our approach does not require a second generative model to predict occupancy as done in structured latent-based approaches (Xiang et al., 2025; He et al., 2025; Wu et al., 2025), simplifying the overall pipeline.

## 3.5 GENERATIVE MODEL

To demonstrate our latent representation, we train a flow-matching model that generates 3D latents conditioned on an image of an object. We rely on a standard Diffusion Transformer (DiT) architecture (Peebles & Xie, 2023), with a zero-initialized learnable positional encoding for each latent token. The input image is encoded by DINOv2-large image embeddings (Oquab et al., 2024) and a learnable patchification layer. While we originally considered using more explicit camera geometry encoding, *e.g.*, Plucker ray embeddings, we found in practice that such an approach reduced overall performance (see Tab. S6 for an ablation). In total, the model has 623 million parameters (see Fig. S7).

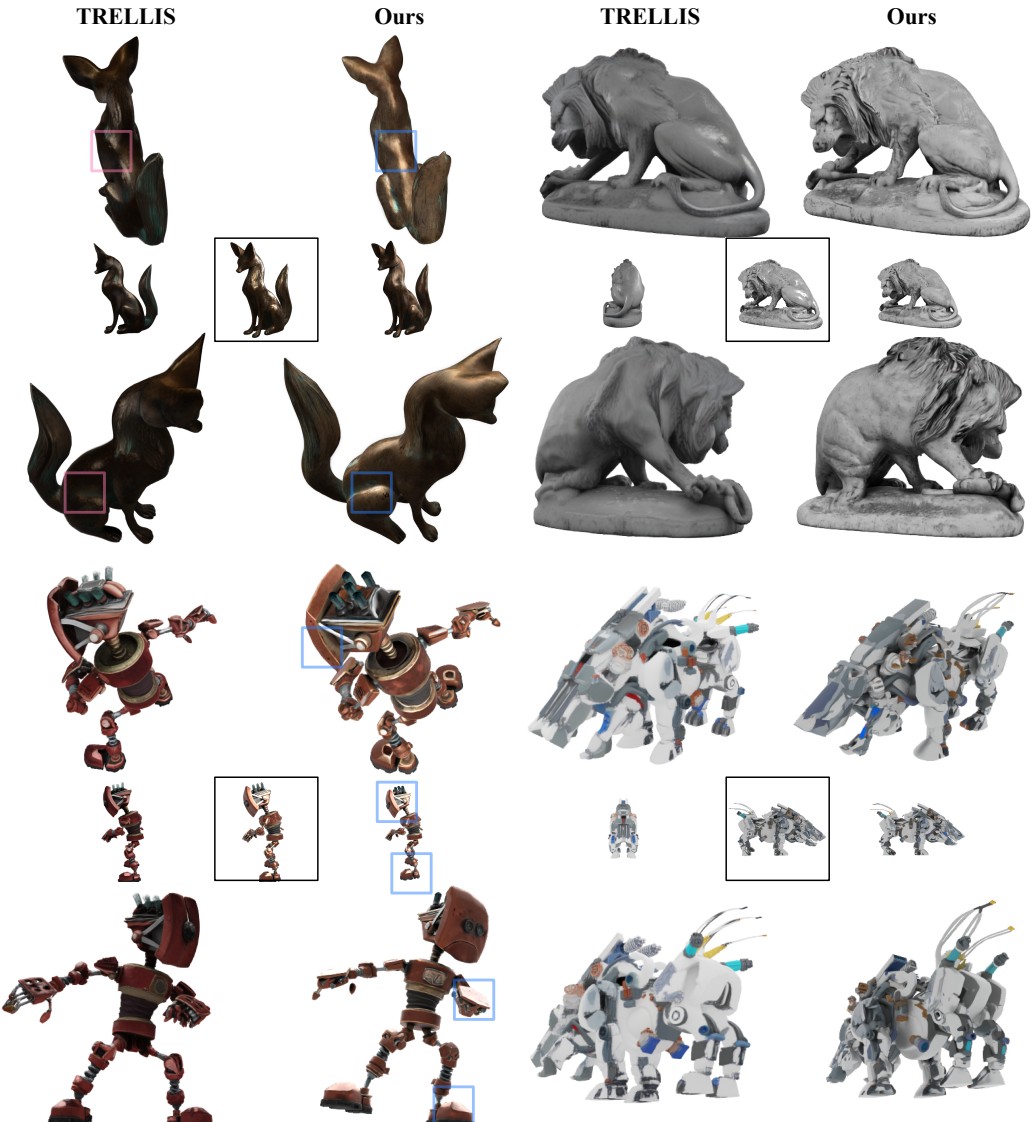

Figure 5: **Single image to 3D results.** The input image is shown at the center of each set with black border. The rendering at the input view is shown with the input image. Please refer to Tab. 3 for quantitative results. Mesh credit: Eleanie (2025); Rigsters (2017); 3d-coat (2015); 3Dji (2025).

For each training sample, we rotate the world coordinate system so the input view's camera pose is set to the identity orientation, removing the need for the model to infer 3D orientation. As a result, the trained model's outputs align with the input view at identity orientation. We train the model for 600k iterations on the tokenizer-training set (effective batch size 256 on 128 H100 GPUs for 20 days).

# 4 EXPERIMENTS

We first train the latent representation, and once learned, we then train a latent flow-matching model conditioned on an input image. See Sec. E for more implementation details. We discuss the training and evaluation of our latent representation in Sec. 4.1, and our image-to-3d model in Sec. 4.2.

## 4.1 RECONSTRUCTION

**Datasets.** We train the encoder-decoder on the 500k high-quality object subset of Objaverse-XL (Deitke et al., 2023) as selected by TRELLIS (Xiang et al., 2025). Unlike TRELLIS, instead

Table 1: **Reconstruction on Toys4k.** We provide input needed by individual methods. TRELLIS (Xiang et al., 2025) takes the ground-truth mesh and 150 sphere-distributed renderings. Ours uses RGBD images from 150 evenly distributed views. For appearance evaluation, we render each model's output from 100 random cameras, varying difficulty by adjusting camera radius. Please refer to Fig. 4 for qualitative results and Sec. C for comprehensive quantitative results. The better one is highlighted.

| Method | Simple, Camera Radius [3, 4] | | | Hard, Camera Radius [1, 3] | | |
|---|---|---|---|---|---|---|
| | PSNR↑ | SSIM↑ | LPIPS↓ | PSNR↑ | SSIM↑ | LPIPS↓ |
| TRELLIS | $31.12\pm3.39$ | $0.974\pm0.022$ | $0.034\pm0.022$ | $27.57\pm3.38$ | $0.941\pm0.050$ | $0.090\pm0.055$ |
| Ours | $34.16\pm3.39$ | $0.985\pm0.016$ | $0.023\pm0.018$ | $32.36\pm3.77$ | $0.967\pm0.040$ | $0.055\pm0.046$ |

Table 2: **Geometric reconstruction evaluation**. We report Chamfer distances multiplied by $10^4$ for readability, computed using 100k sampled points each from ground-truth and reconstruction. As 3DTopia-XL (Chen et al., 2025b) and TripoSG (Li et al., 2025a) can be sensitive to input geometry, we also list variants with their 10% worst-performing objects removed. We separate our tested approaches based on those that require ground-truth coarse geometry for decoding the latent representation, and those that do not utilize this information. Our method outputs the best geometry among the approaches in the latter category, and it is competitive with the techniques in the former while using a 10x smaller latent space. Best and 2nd-Best methods in each category are highlighted.

| | Method | Appearance | Latent size | PBR-Objaverse | Toys4k | GSO |
|---|---|---|---|---|---|---|
| 0 | GT | – | – | $82.49\pm21.39$ | $76.03\pm24.30$ | $87.50\pm21.04$ |
| | **Requires coarse geometry oracle:** | | | | | |
| 1 | TripoSF (He et al., 2025) | ✗ | $\approx 244k \times 11$ | $83.11\pm21.45$ | $76.99\pm24.95$ | $87.63\pm21.67$ |
| 2 | TRELLIS (Xiang et al., 2025) | ✓ | $\approx 20k \times 11$ | $95.16\pm20.77$ | $92.21\pm25.99$ | $105.5\pm22.44$ |
| 3-1 | 3DTopia-XL (Chen et al., 2025b) | ✓ | $2048 \times 64$ | $412.5\pm1129.$ | $153.9\pm305.8$ | $98.50\pm19.66$ |
| 3-2 | (worst 10% removed) | ✓ | $2048 \times 64$ | $135.4\pm95.75$ | $90.61\pm23.96$ | $93.62\pm12.86$ |
| 4 | Ours (oracle, mesh decoder) | ✓ | $8192 \times 32$ | $87.02\pm24.19$ | $80.32\pm27.30$ | $94.87\pm23.42$ |
| | **Does not utilize coarse geometry oracle:** | | | | | |
| 5-1 | TripoSG (Li et al., 2025a) | ✗ | $2048 \times 64$ | $269.2\pm260.0$ | $299.7\pm265.9$ | $301.3\pm300.2$ |
| 5-2 | (worst 10% removed) | ✗ | $2048 \times 64$ | $199.6\pm125.3$ | $230.5\pm162.3$ | $219.7\pm173.6$ |
| 6 | Shape Tokens (Chang et al., 2024) | ✗ | $1024 \times 16$ | $126.0\pm23.20$ | $119.8\pm28.02$ | $130.5\pm20.72$ |
| 7-1 | Ours (no mesh decoder) | ✓ | $8192 \times 32$ | $94.08\pm22.01$ | $88.30\pm25.23$ | $98.66\pm21.50$ |
| 7-2 | Ours (mesh decoder) | ✓ | $8192 \times 32$ | $87.17\pm24.29$ | $80.55\pm27.59$ | $95.19\pm23.64$ |

of using all 500k objects for training, we divide the data into training, validation, and test sets in an 8:1:1 ratio. For each object, we pair it with 3 lighting conditions: 1) fixed smooth area lighting (matching TRELLIS)[2], 2) an all-white environment map, and 3) randomly placed lights. For each configuration, we render using Blender from 150 viewpoints uniformly distributed on a sphere, to sample the surface light field as input for our encoder. We render from 100 random viewpoints to supervise our view-dependent Gaussian decoder.

We evaluate the models on Toys4k (Stojanov et al., 2021), GSO (Downs et al., 2022), and Objaverse-XL (Deitke et al., 2023). For Objaverse-XL, we select a subset of 200 objects with PBR materials, which we dub PBR-Objaverse.

**Qualitative results:** Fig. 4 shows a few objects with view-dependent appearance, including specular reflections from metallic surfaces and Fresnel reflections when viewed at grazing angles.

**Quantitative results (appearance):** To evaluate appearance quality, we render the 3DGS from 100 random views on a sphere and measure PSNR, SSIM (Wang et al., 2004), and LPIPS (Zhang et al., 2018). Tab. 1 shows reconstruction metrics under different zoom-in levels on the Toys4k dataset rendered with TRELLIS's training lighting condition. Our surface light-field representation outperforms competitor appearance representations across all the tested metrics. More evaluations on other datasets and lightings are described in Sec. C.

**Quantitative results (geometry):** To evaluate the quality of reconstructed 3D geometry, we estimate ground truth point clouds by unprojecting the rendered depth of a target object from 100 uniformly distributed views on the sphere and randomly selecting 100k reference points. We then compute

---

[2] https://github.com/microsoft/TRELLIS/blob/6b0d64751ad54d9c3/dataset_toolkits/blender_script/render.py#L178-L209

| Input View | Ours (same view) | TRELLIS (same view) |
|:---:|:---:|:---:|

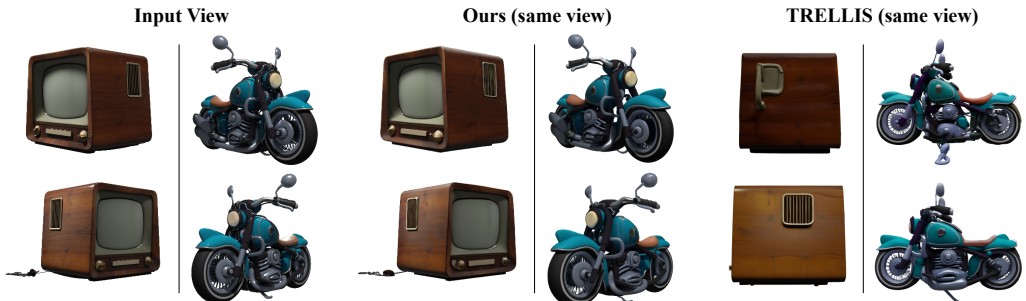

Figure 6: **Fidelity to input view.** Our image-to-3d generative model respects the coordinate system of the input view. In contrast, existing state-of-the-art techniques, *e.g.*, TRELLIS (Xiang et al., 2025), do not. Mesh credit: Virtual Museums of Małopolska (2016); animanyarty (2022).

Chamfer distance in Sec. C.1 between these ground truth point clouds and reconstructed ones. For LiTo and Chang et al. (2024), we sample 100k points from the flow-matching velocity decoder to produce the output point cloud. To fairly compare to baselines that output meshes (Xiang et al., 2025; Li et al., 2025a; He et al., 2025), we also train a mesh decoder (Sec. E.6, Fig. S6, and Fig. S1). Similar to the ground truth points, we unproject rendered depths of the mesh from another set of 100 views on the sphere and select 100k points for the Chamfer calculation.

Tab. 2 shows geometry evaluation when the input is lit with TRELLIS's training lighting. Our method (row 7-1 and 7-2) outperforms most geometry-only latent representations, despite we additionally represent appearance information and do not utilize additional ground truth coarse geometry information that other state-of-the-art approaches require (Xiang et al., 2025; He et al., 2025).

Table 3: **Single-image-to-3D generation on Toys4k.** KID is reported by $\times 100$. CFG scale for both models are 3.0. The best is highlighted. See Fig. 5, 6 for qualitative results.

| Method | CLIP↑ | Conditioning View | | | | Novel View | | | |
|---|---|---|---|---|---|---|---|---|---|
| | | FID↓ | KID↓ | FID$_{dino}$↓ | KID$_{dino}$↓ | FID↓ | KID↓ | FID$_{dino}$↓ | KID$_{dino}$↓ |
| TRELLIS | 0.899±0.045 | 12.84 | 0.088 | 84.692 | 2.311 | 7.600 | 0.100 | 67.458 | 3.166 |
| Ours | 0.905±0.041 | 6.219 | 0.009 | 41.621 | 1.333 | 6.216 | 0.058 | 66.530 | 3.522 |

## 4.2 GENERATION

Fig. 5 contains qualitative results. Our model generates complex geometry and view-dependent appearance, despite being trained on other lighting types. We also visualize our model's input view fidelity compared to TRELLIS in Fig. 6 to verify our training strategy in Sec. 3.5.

We quantitatively evaluate generation results with the same fixed area lighting as TRELLIS to allow a fair comparison. We calculate two distribution-wise metrics. First, to evaluate the fidelity of the generative model to the input content, we render the generated 3D asset at the same pose as the conditioning view. As shown in Tab. 3, our approach produces significantly improved FID (Heusel et al., 2017) and KID (Binkowski et al., 2018) scores in this setting compared to TRELLIS. Second, to measure the overall quality of the generated asset, we render from four novel views distributed around the object at a pitch of $30°$, following the evaluation setup of TRELLIS (Xiang et al., 2025). As shown in Tab. 3, despite our model's increased faithfulness to the input view, the overall generation performance does not significantly degrade. Please refer to Sec. D for more studies.

## 5 CONCLUSION

We propose an autoencoder that learns a compact latent space for 3D assets with view-dependent appearance. In particular, we build an encoding of the surface light field, that can be easily produced via multi-view RGBD rendering. With a flow-matching geometry decoder (or a separately-trained mesh decoder) and a view-dependent Gaussian decoder, our representation can be easily applied with an off-the-shelf DiT for generating view-dependent 3D assets. We validate the performance of our view-dependent 3D representation in both reconstruction and generation.

ACKNOWLEDGEMENTS

We thank Muhammed Kocabas for creating the LiTo demo. We are grateful to Miguel Angel Bautista Martin, Hadi Pouransari, Josh Susskind, Barry Theobald, Yuyang Wang, and the reviewers for their valuable feedback on our paper. We also thank Denise Hui, David Koski, and the broader Apple infrastructure team for maintaining the computing resources that supported this work. Names are listed in alphabetical order by last name.

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

APPENDIX – LITO: SURFACE LIGHT FIELD TOKENIZATION

This supplement is organized as follows:

1. Sec. A discusses more on related works;
2. Sec. B discusses limitations;
3. Sec. C provides more comprehensive reconstruction quantitative results;
4. Sec. D provides more comprehensive generation quantitative results;
5. Sec. E introduces more implementation details;
6. Sec. F showcases more studies;
7. Sec. G breaks down full contributions.

## A   MORE RELATED WORKS

Tab. S1 provides an overview of related works with respect to 1) how they model the geometry; 2) how they model the appearance; 3) the requirements on the data preparation to enable the model training; 4) the compactness of the latent size; 5) the input to the encoder and the training dataset.

## B   LIMITATIONS

We utilize 3D Gaussians with spherical harmonics to model surface light field. While we show that the improved reconstruction quality as we increase the degree of the spherical harmonics, we are constraint by the 3DGS implementation that supports up to degree 3, which limits our capability to faithfully reconstruct transparent or high-frequency specularities.

## C   COMPREHENSIVE RECONSTRUCTION RESULTS

We provide comprehensive quantitative results for reconstruction in Tab. S2, S3, and S4. As discussed in Sec. 4.1, we pair each dataset with three distinct lighting conditions to thoroughly evaluate the appearance modeling capabilities of our method. Unlike previous approaches, which primarily assess performance on zoomed-out views, we additionally evaluate appearance modeling under close-up settings. Close-up views demand greater fidelity in capturing high-frequency details, where all methods face challenges; nevertheless, LiTo consistently demonstrates the most robust performance.

Further, we provide qualitative results for reconstructed mesh in Fig. S1.

### C.1   METRICS

We use the following definition for Chamfer distance for any specific 3D asset reported in the quantitative results:

$$\text{CD}(\mathcal{X}_{\text{GT}}, \mathcal{X}_{\text{pred}}) = \frac{1}{|\mathcal{X}_{\text{GT}}|} \sum_{\mathbf{x}_{\text{GT}} \in \mathcal{X}_{\text{GT}}} \min_{\mathbf{x}_{\text{pred}} \in \mathcal{X}_{\text{pred}}} \|\mathbf{x}_{\text{GT}} - \mathbf{x}_{\text{pred}}\|_2$$
$$+ \frac{1}{|\mathcal{X}_{\text{pred}}|} \sum_{\mathbf{x}_{\text{pred}} \in \mathcal{X}_{\text{pred}}} \min_{\mathbf{x}_{\text{GT}} \in \mathcal{X}_{\text{GT}}} \|\mathbf{x}_{\text{GT}} - \mathbf{x}_{\text{pred}}\|_2, \tag{S1}$$

where $\mathcal{X}_{\text{GT}}$ and $\mathcal{X}_{\text{pred}}$ denote the ground-truth and predicted sets of points respectively.

### C.2   ABLATIONS ON MODEL DESIGNS

As far as we know (see Tab. S1), we are the first to utilize 1) viewing directions in the encoder; and 2) higher order spherical harmonics in the decoder during 3D asset tokenization training. Thus, we are mainly interested in understanding the effects of these design choices.

Table S1: **Recent latent 3D representations.** The table provides a summary of recent 3D representations and their properties. We compare the properties that are relevant to machine learning applications. *Minimal preprocessing* indicates how easy is it to utilize a 3D dataset (*e.g.*, do we need to convert data to watertight meshes, do we need optimization radiance fields to acquire the actual training dataset). *Continuous latent* indicates whether the 3D representation is fully differentiable (*e.g.*, no graph topology or sparsity patterns). Total latent dimension indicates the total size to represent one scene. Note that there may be multiple variants of the same method with different latent dimensions. We choose the representative one in each paper. * indicates a second generative model is used in the paper to add texture to a texture-less meshes.

| name | geometry | appearance | data requirements | total latent dimension | input to encoder | training dataset |
|---|---|---|---|---|---|---|
| DDPM-PointCloud (2021) | p(xyz) | - | point cloud | 256 | point cloud ($\mathbf{x}$) | ShapeNet |
| PointFlow (2019) | p(xyz) | - | point cloud | 512 | point cloud ($\mathbf{x}$) | ShapeNet |
| ShapeGF (2020) | p(xyz) | - | point cloud | 256 | point cloud ($\mathbf{x}$) | ShapeNet |
| Shape Token (2024) | p(xyz) | - | point cloud | $1024 \times 16$ | point cloud ($\mathbf{x}$) | Objaverse |
| **Ours** | p(xyz) | view-dep. 3DGS | multiview RGBD | $8192 \times 32$ | surface light field ($\mathbf{x}, \mathbf{c}, \mathbf{d}$) | Objaverse, ObjaverseXL |
| Point-E (2022) | fixed size point set | diffuse RGB | point cloud ($\mathbf{x}$) | - | - | proprietary dataset |
| LION (2022) | fixed size point set | - | point cloud | $128 + 8192$ | point cloud ($\mathbf{x}$) | ShapeNet |
| 3DShape2VecSet (2023) | occupancy field | - | watertight mesh | $512 \times 32$ | point cloud ($\mathbf{x}$) | ShapeNet-watertight |
| 3DILG (2022) | occupancy field | - | watertight mesh | $512 \times 2$ | point cloud ($\mathbf{x}$) | ShapeNet-watertight |
| Michelangelo (2023) | occupancy field | - | watertight mesh | $512 \times 64 + 768$ | point cloud ($\mathbf{x}, \hat{\mathbf{n}}$) | ShapeNet, 3D cartoon monster |
| CLAY (2024) | occupancy field | -* | watertight mesh | $2048 \times 64$ | point cloud ($\mathbf{x}$) | Objaverse |
| Dora (2025a) | occupancy field | - | watertight mesh | $1280 \times 64$ | point cloud ($\mathbf{x}$) | Objaverse |
| Pandora3D (2025) | occupancy field | -* | watertight mesh | $2048 \times 64$ | point cloud ($\mathbf{x}, \hat{\mathbf{n}}$) | Objaverse, ObjaverseXL, ABO, BuildingNet, HSSD, Toy4k, polygone dataset, proprietary |
| Direct3D (2024) | occupancy grid | - | watertight mesh | $3 \times 32 \times 32 \times 16$ | point cloud ($\mathbf{x}, \hat{\mathbf{n}}$) | proprietary dataset |
| Direct3D-s2 (2025) | SDF grid | - | watertight mesh | $(128^3 \times 16)$ | point cloud ($\mathbf{x}, \hat{\mathbf{n}}$) | Objaverse, ObjaverseXL |
| XCube (2024) | occupancy grid | - | watertight mesh | $16^3 \times 16 +$ more | occupancy grid | ShapeNet, Objaverse |
| LT3SD (2025) | UDF grid | - | watertight mesh | $(2 \times 1 \times 2) \times (5 + 4^3 \times 4 + 16^3 \times 4)$ | UDF grid | 3D Front |
| Diffusion-SDF (2023) | SDF field | - | watertight mesh | 768 | point cloud ($\mathbf{x}$) | ShapeNet-watertight, YCB |
| MOSAIC-SDF (2024) | SDF field | - | watertight mesh and optimization | $1024 \times (3 + 1 + 7^3)$ | - | ShapeNet-watertight, scalable 3D captioning dataset |
| TripoSG (2025a) | SDF field | - | watertight mesh | $2048 \times 64$ | point cloud ($\mathbf{x}, \hat{\mathbf{n}}$) | Objaverse, ObjaverseXL |
| Hunyuan3D 2.0 (2025) | SDF field | -* | watertight mesh | $3072 \times 64$ | point cloud ($\mathbf{x}$) | Objaverse, ObjaverseXL, more |
| Make-A-Shape (2024) | SDF grid | - | watertight mesh | 9M | - | 18 datasets |
| 3DTopia-XL (2025b) | PrimX (SDF field) | RGB, PBR | PrimX optimization | $2048 \times (3 + 1 + 4^3)$ $= 139,264$ | PrimX | Objaverse |
| Sparc3D (2025b) | SDF grid | - | watertight mesh, grid optimization | unknown | SDF grid | |
| Volume Diffusion (2023) | radiance field | diffuse RGB | run inference network | $32^3 \times 4$ | multiview images | Objaverse |
| TRELLIS (2025) | occupancy grid | diffuse 3DGS | multiview DINOv2 | $\sim$20,000 $\times 11$ ($64^3$ grid) | sparse feature grid | Objaverse, ObjaverseXL, ABO, 3D-future, HSSD |
| TripoSF (2025) | SDF grid | - | multiview depth and normal | $\sim$183,000 $\times 11$ ($256^3$ grid) | point cloud ($\mathbf{x}, \hat{\mathbf{n}}$) | Objaverse, ObjaverseXL |

When examining LPIPS across Tab. S2, S3, and S4, we observe: 1) increasing the degree of spherical harmonics from 0 to 3 improves the capacity consistently, *e.g.*, from row 1-3 to 1-6 (or row 2-3 to 2-6, 3-3 to 3-6) in all three tables; and 2) simply adding ray information does not directly enhance appearance modeling performance, *e.g.*, row 1-2 *vs.* 1-3 (or row 2-2 *vs.* 2-3, 3-2 *vs.* 3-3). We hypothesize that this is because zero-degree spherical harmonics cannot capture view-dependent effects, which then becomes a bottleneck, preventing the model from fully leveraging the information contained in the view directions. To verify, we ablate by removing the ray information from our encoder when using 3-degree spherical harmonics. The improvement in row 1-6, which incorporates ray information, from 1-7 (or row 2-6 *vs.* 2-7, 3-6 *vs.* 3-7) corroborates our hypothesis.

## C.3 ABLATIONS ON NUMBER OF INPUT VIEWS IN INFERENCE

We are interested in understanding to what extent our approach is robust to the discrepancies between the number of input views during training and inference. Quantitative evaluations are in Tab. S5.

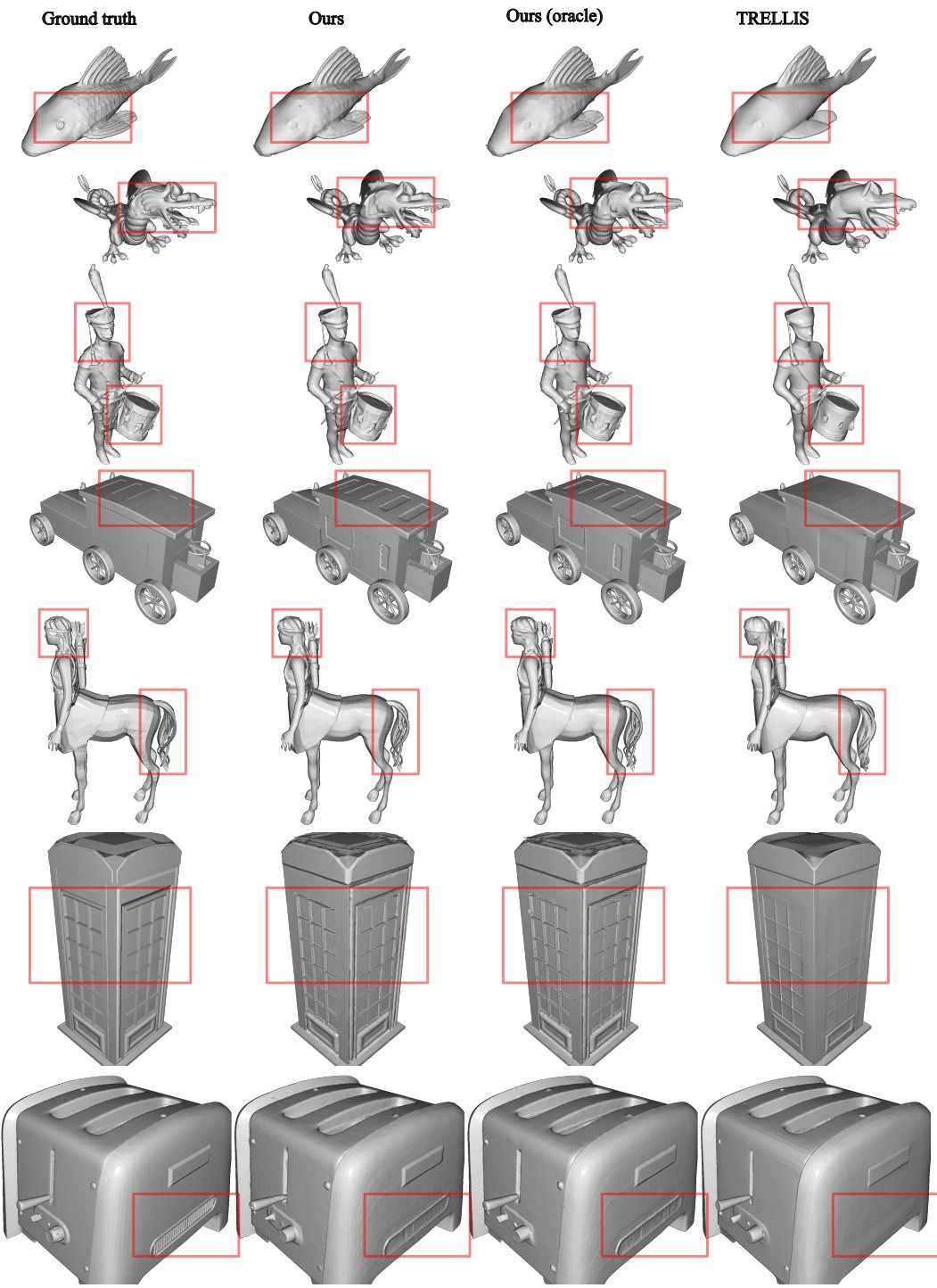

Figure S1: **Mesh comparisons.** We demonstrate the qualities of our mesh decoder results to TRELLIS. As highlighted, our produced mesh maintains more details. Mesh credit: alzarac (2019); Alienor.org, Conseil des musées (2016); 1812panorama (2019); AdamJonesCGD (2020); nastasyas (2019); a108082046 (2022); GJ2012 (2013).

Table S2: **Reconstruction on Toys4k.** For 3D assets, we adapt inputs per model. TRELLIS (Xiang et al., 2025) takes the ground-truth mesh and 150 sphere-distributed renderings. Ours uses RGBD images from 150 evenly distributed views. For appearance evaluation, we render each model's output from 100 random cameras, varying difficulty by adjusting camera radius. Each model is further evaluated under three distinct lighting conditions. Importantly, no separate models are trained; all evaluations are conducted on the same model. As a result, we conduct evaluations at the scale of over 3000 (objects) $\times$ 100 (views) $\times$ 2 (difficulties) $\times$ 3 (lightings) $\approx$ **1.8 million images**. We report Chamfer distances multiplied by $10^4$ for readability, computed using 100k sampled points each from the ground truth and reconstruction. We report in the format of mean±std, where the standard deviation is computed across objects.

| | Method | SH Deg | Enc Ray | Pred Occ | Mesh | Simple, Camera Radius [3, 4] | | | Hard, Camera Radius [1, 3] | | | CD (100k)↓ |
|---|---|---|---|---|---|---|---|---|---|---|---|---|
| | | | | | | PSNR↑ | SSIM↑ | LPIPS↓ | PSNR↑ | SSIM↑ | LPIPS↓ | |
| | | | | | | **Uniform Lighting** | | | | | | |
| 1-1 | TRELLIS | 0 | ✗ | - | ✓ | 28.17±4.09 | 0.970±0.024 | 0.039±0.024 | 24.63±4.01 | 0.934±0.054 | 0.098±0.059 | 95.19±26.41 |
| 1-2 | Ours | 0 | ✗ | ✗ | ✗ | 34.40±3.62 | 0.984±0.017 | 0.025±0.019 | 32.19±3.95 | 0.965±0.042 | 0.059±0.047 | 90.22±25.48 |
| 1-3 | Ours | 0 | ✓ | ✗ | ✗ | 34.44±3.47 | 0.984±0.017 | 0.026±0.020 | 32.18±3.79 | 0.964±0.042 | 0.060±0.048 | 89.24±25.34 |
| 1-4 | Ours | 1 | ✓ | ✗ | ✗ | 35.12±3.39 | 0.986±0.015 | 0.023±0.017 | 33.17±3.76 | 0.968±0.040 | 0.054±0.044 | 88.42±25.21 |
| 1-5 | Ours | 2 | ✓ | ✗ | ✗ | 35.32±3.45 | 0.986±0.016 | 0.023±0.017 | 33.29±3.80 | 0.969±0.040 | 0.055±0.044 | 88.94±25.34 |
| 1-6 | Ours | 3 | ✓ | ✗ | ✗ | 35.32±3.38 | 0.986±0.015 | 0.022±0.017 | 33.39±3.73 | 0.969±0.039 | 0.053±0.044 | 88.13±25.30 |
| 1-7 | Ours | 3 | ✗ | ✗ | ✗ | 35.54±3.63 | 0.986±0.015 | 0.023±0.017 | 33.37±3.97 | 0.969±0.040 | 0.055±0.044 | 89.63±25.16 |
| 1-8 | Ours | 3 | ✓ | ✓ | ✗ | 35.27±3.36 | 0.986±0.015 | 0.022±0.017 | 33.38±3.71 | 0.969±0.040 | 0.052±0.044 | 88.13±25.29 |
| 1-9 | Ours | 3 | ✓ | ✓ | ✓ | 35.27±3.36 | 0.986±0.015 | 0.022±0.017 | 33.38±3.71 | 0.969±0.040 | 0.052±0.044 | 80.42±27.90 |
| 1-10 | Oracle | 3 | – | – | ✓ | 35.26±3.34 | 0.986±0.015 | 0.022±0.017 | 33.42±3.69 | 0.970±0.039 | 0.051±0.043 | 80.17±27.58 |
| | | | | | | **TRELLIS Lighting** | | | | | | |
| 2-1 | TRELLIS | 0 | ✗ | - | ✓ | 31.12±3.39 | 0.974±0.022 | 0.034±0.022 | 27.57±3.38 | 0.941±0.050 | 0.090±0.055 | 92.21±25.99 |
| 2-2 | Ours | 0 | ✗ | ✗ | ✗ | 32.47±3.83 | 0.980±0.020 | 0.029±0.022 | 30.21±4.19 | 0.958±0.046 | 0.067±0.053 | 90.00±25.39 |
| 2-3 | Ours | 0 | ✓ | ✗ | ✗ | 32.47±3.69 | 0.980±0.020 | 0.029±0.022 | 30.21±4.06 | 0.957±0.046 | 0.068±0.052 | 89.12±25.38 |
| 2-4 | Ours | 1 | ✓ | ✗ | ✗ | 34.00±3.38 | 0.984±0.016 | 0.025±0.019 | 32.03±3.74 | 0.965±0.040 | 0.059±0.047 | 88.46±25.13 |
| 2-5 | Ours | 2 | ✓ | ✗ | ✗ | 34.06±3.40 | 0.984±0.016 | 0.024±0.019 | 32.12±3.79 | 0.966±0.041 | 0.058±0.047 | 88.82±25.30 |
| 2-6 | Ours | 3 | ✓ | ✗ | ✗ | 34.19±3.39 | 0.985±0.016 | 0.024±0.019 | 32.36±3.77 | 0.967±0.040 | 0.056±0.046 | 88.30±25.23 |
| 2-7 | Ours | 3 | ✗ | ✗ | ✗ | 34.16±3.68 | 0.985±0.017 | 0.025±0.019 | 32.11±4.04 | 0.966±0.041 | 0.058±0.047 | 89.35±25.12 |
| 2-8 | Ours | 3 | ✓ | ✓ | ✗ | 34.16±3.39 | 0.985±0.016 | 0.023±0.018 | 32.36±3.77 | 0.967±0.040 | 0.055±0.046 | 88.30±25.23 |
| 2-9 | Ours | 3 | ✓ | ✓ | ✓ | 34.16±3.39 | 0.985±0.016 | 0.023±0.018 | 32.36±3.77 | 0.967±0.040 | 0.055±0.046 | 80.55±27.59 |
| 2-10 | Oracle | 3 | – | – | ✓ | 34.14±3.37 | 0.985±0.016 | 0.023±0.018 | 32.38±3.74 | 0.967±0.040 | 0.054±0.045 | 80.32±27.30 |
| | | | | | | **Random Lighting** | | | | | | |
| 3-1 | TRELLIS | 0 | ✗ | - | ✓ | 27.94±3.77 | 0.966±0.025 | 0.038±0.024 | 24.37±3.66 | 0.927±0.054 | 0.098±0.058 | 93.95±25.89 |
| 3-2 | Ours | 0 | ✗ | ✗ | ✗ | 32.12±3.23 | 0.981±0.018 | 0.026±0.021 | 30.08±3.67 | 0.961±0.043 | 0.062±0.051 | 90.15±25.59 |
| 3-3 | Ours | 0 | ✓ | ✗ | ✗ | 32.18±3.12 | 0.981±0.019 | 0.026±0.021 | 30.11±3.57 | 0.960±0.044 | 0.063±0.052 | 89.45±25.36 |
| 3-4 | Ours | 1 | ✓ | ✗ | ✗ | 33.02±2.92 | 0.984±0.017 | 0.023±0.019 | 31.20±3.39 | 0.965±0.041 | 0.057±0.047 | 88.65±25.24 |
| 3-5 | Ours | 2 | ✓ | ✗ | ✗ | 33.13±2.99 | 0.984±0.017 | 0.023±0.019 | 31.34±3.49 | 0.966±0.041 | 0.058±0.048 | 89.18±25.43 |
| 3-6 | Ours | 3 | ✓ | ✗ | ✗ | 33.22±2.95 | 0.984±0.017 | 0.023±0.019 | 31.50±3.41 | 0.966±0.041 | 0.056±0.048 | 88.30±25.31 |
| 3-7 | Ours | 3 | ✗ | ✗ | ✗ | 33.23±3.32 | 0.984±0.017 | 0.024±0.019 | 31.30±3.83 | 0.965±0.041 | 0.058±0.049 | 89.69±25.19 |
| 3-8 | Ours | 3 | ✓ | ✓ | ✗ | 33.18±2.93 | 0.984±0.017 | 0.022±0.019 | 31.49±3.39 | 0.966±0.041 | 0.055±0.048 | 88.31±25.32 |
| 3-9 | Ours | 3 | ✓ | ✓ | ✓ | 33.18±2.92 | 0.984±0.017 | 0.022±0.019 | 31.50±3.39 | 0.966±0.041 | 0.055±0.047 | 80.39±27.95 |
| 3-10 | Oracle | 3 | – | – | ✓ | 33.15±2.90 | 0.984±0.016 | 0.022±0.019 | 31.50±3.36 | 0.967±0.040 | 0.054±0.047 | 80.11±27.51 |

# D COMPREHENSIVE GENERATION RESULTS

As demonstrated in Fig. 6, we are interested in aligning the generation with the input view faithfully. To achieve this, for each sample used during training, we carefully rotate the world coordinate system such that the input view's corresponding camera poses are at the identity orientation. This relieves the model from the burden of inferring the orientation of 3D space during training. Further, we consider utilizing the view direction during the generative model training as well to enable the model be aware of 3D orientation. Since we make the orientation identity, ray information essentially means the availability of camera intrinsics. Then, during inference, we use an off-the-shelf intrinsic estimator (Wang et al., 2025) to obtain the intrinsics. However, as shown in row 3 *vs.* 2 in Tab. S6, it seems like the intrinsic information is unnecessary. Thus we use the generative model trained without any ray information to report our qualitative and quantitative results in the paper.

Table S3: **Reconstruction on GSO.** For 3D assets, we adapt inputs per model. TRELLIS (Xiang et al., 2025) takes the ground-truth mesh and 150 sphere-distributed renderings. Ours uses RGBD images from 150 evenly distributed views. For appearance evaluation, we render each model's output from 100 random cameras, varying difficulty by adjusting camera radius. Each model is further evaluated under three distinct lighting conditions. Importantly, no separate models are trained; all evaluations are conducted on the same model. As a result, we conduct evaluations at the scale of over 1000 (objects) $\times$ 100 (views) $\times$ 2 (difficulties) $\times$ 3 (lightings) $\approx$ **600 thousand images**. We report Chamfer distances multiplied by $10^4$ for readability, computed using 100k sampled points each from the ground truth and reconstruction. We report in the format of mean±std, where the standard deviation is computed across objects.

| | Method | SH Deg | Enc Ray | Pred Occ | Mesh | Simple, Camera Radius [3, 4] | | | Hard, Camera Radius [1, 3] | | | CD (100k)↓ |
|---|---|---|---|---|---|---|---|---|---|---|---|---|
| | | | | | | PSNR↑ | SSIM↑ | LPIPS↓ | PSNR↑ | SSIM↑ | LPIPS↓ | |
| | | | | | | **Uniform Lighting** | | | | | | |
| 1-1 | TRELLIS | 0 | ✗ | - | ✓ | 27.34±3.82 | 0.947±0.036 | 0.053±0.029 | 23.72±3.66 | 0.883±0.068 | 0.139±0.065 | 108.2±22.56 |
| 1-2 | Ours | 0 | ✗ | ✗ | ✗ | 34.27±3.25 | 0.975±0.022 | 0.034±0.025 | 31.39±3.61 | 0.937±0.046 | 0.093±0.055 | 101.4±22.21 |
| 1-3 | Ours | 0 | ✓ | ✗ | ✗ | 34.04±3.23 | 0.974±0.022 | 0.034±0.025 | 31.15±3.59 | 0.935±0.048 | 0.093±0.055 | 100.0±21.41 |
| 1-4 | Ours | 1 | ✓ | ✗ | ✗ | 34.55±3.18 | 0.976±0.021 | 0.031±0.023 | 31.75±3.60 | 0.939±0.045 | 0.087±0.053 | 99.21±21.44 |
| 1-5 | Ours | 2 | ✓ | ✗ | ✗ | 34.62±3.24 | 0.976±0.021 | 0.031±0.024 | 31.77±3.65 | 0.939±0.046 | 0.087±0.053 | 99.68±21.75 |
| 1-6 | Ours | 3 | ✓ | ✗ | ✗ | 34.69±3.22 | 0.976±0.021 | 0.031±0.024 | 31.88±3.65 | 0.940±0.046 | 0.086±0.053 | 98.91±21.65 |
| 1-7 | Ours | 3 | ✗ | ✗ | ✗ | 34.93±3.24 | 0.977±0.020 | 0.031±0.023 | 32.00±3.63 | 0.942±0.044 | 0.087±0.053 | 101.0±22.20 |
| 1-8 | Ours | 3 | ✓ | ✓ | ✗ | 34.67±3.21 | 0.976±0.021 | 0.031±0.024 | 31.88±3.65 | 0.940±0.046 | 0.086±0.053 | 98.92±21.67 |
| 1-9 | Ours | 3 | ✓ | ✓ | ✓ | 34.67±3.21 | 0.976±0.021 | 0.031±0.024 | 31.88±3.65 | 0.940±0.046 | 0.086±0.053 | 92.95±24.19 |
| 1-10 | Oracle | 3 | – | – | ✓ | 34.66±3.20 | 0.976±0.021 | 0.030±0.023 | 31.92±3.65 | 0.941±0.045 | 0.085±0.053 | 92.70±24.05 |
| | | | | | | **TRELLIS Lighting** | | | | | | |
| 2-1 | TRELLIS | 0 | ✗ | - | ✓ | 30.81±2.67 | 0.958±0.028 | 0.047±0.026 | 27.21±2.56 | 0.907±0.055 | 0.126±0.058 | 105.5±22.44 |
| 2-2 | Ours | 0 | ✗ | ✗ | ✗ | 33.99±2.54 | 0.978±0.017 | 0.033±0.023 | 31.65±2.71 | 0.948±0.036 | 0.089±0.052 | 101.2±21.96 |
| 2-3 | Ours | 0 | ✓ | ✗ | ✗ | 33.71±2.43 | 0.978±0.018 | 0.033±0.024 | 31.40±2.62 | 0.947±0.037 | 0.088±0.051 | 99.62±21.28 |
| 2-4 | Ours | 1 | ✓ | ✗ | ✗ | 34.75±2.60 | 0.980±0.016 | 0.030±0.022 | 32.50±2.87 | 0.952±0.035 | 0.080±0.048 | 98.93±21.33 |
| 2-5 | Ours | 2 | ✓ | ✗ | ✗ | 34.87±2.68 | 0.980±0.017 | 0.030±0.022 | 32.58±2.95 | 0.952±0.036 | 0.081±0.049 | 99.28±21.58 |
| 2-6 | Ours | 3 | ✓ | ✗ | ✗ | 34.91±2.65 | 0.980±0.016 | 0.029±0.022 | 32.67±2.95 | 0.952±0.036 | 0.080±0.049 | 98.66±21.50 |
| 2-7 | Ours | 3 | ✗ | ✗ | ✗ | 35.19±2.72 | 0.981±0.016 | 0.030±0.022 | 32.79±2.97 | 0.953±0.034 | 0.081±0.049 | 100.6±21.99 |
| 2-8 | Ours | 3 | ✓ | ✓ | ✗ | 34.89±2.64 | 0.980±0.016 | 0.029±0.022 | 32.68±2.94 | 0.952±0.036 | 0.079±0.049 | 98.64±21.49 |
| 2-9 | Ours | 3 | ✓ | ✓ | ✓ | 34.89±2.64 | 0.980±0.016 | 0.029±0.022 | 32.68±2.94 | 0.952±0.036 | 0.079±0.049 | 95.19±23.64 |
| 2-10 | Oracle | 3 | – | – | ✓ | 34.87±2.63 | 0.981±0.016 | 0.029±0.021 | 32.70±2.94 | 0.953±0.036 | 0.078±0.048 | 94.87±23.42 |
| | | | | | | **Random Lighting** | | | | | | |
| 3-1 | TRELLIS | 0 | ✗ | - | ✓ | 27.66±3.26 | 0.948±0.033 | 0.050±0.028 | 24.11±3.08 | 0.886±0.064 | 0.133±0.062 | 107.5±22.36 |
| 3-2 | Ours | 0 | ✗ | ✗ | ✗ | 33.09±2.47 | 0.977±0.018 | 0.031±0.023 | 30.97±2.81 | 0.945±0.039 | 0.086±0.052 | 101.3±22.24 |
| 3-3 | Ours | 0 | ✓ | ✗ | ✗ | 32.97±2.40 | 0.976±0.018 | 0.031±0.023 | 30.82±2.77 | 0.943±0.040 | 0.087±0.052 | 100.0±21.43 |
| 3-4 | Ours | 1 | ✓ | ✗ | ✗ | 33.46±2.41 | 0.978±0.017 | 0.028±0.021 | 31.41±2.81 | 0.947±0.038 | 0.080±0.049 | 99.29±21.44 |
| 3-5 | Ours | 2 | ✓ | ✗ | ✗ | 33.61±2.47 | 0.978±0.017 | 0.029±0.022 | 31.55±2.88 | 0.947±0.038 | 0.081±0.049 | 99.67±21.81 |
| 3-6 | Ours | 3 | ✓ | ✗ | ✗ | 33.67±2.46 | 0.979±0.017 | 0.028±0.022 | 31.65±2.89 | 0.948±0.038 | 0.080±0.050 | 98.93±21.68 |
| 3-7 | Ours | 3 | ✗ | ✗ | ✗ | 33.98±2.53 | 0.980±0.016 | 0.028±0.021 | 31.84±2.93 | 0.949±0.036 | 0.081±0.049 | 100.9±22.27 |
| 3-8 | Ours | 3 | ✓ | ✓ | ✗ | 33.64±2.43 | 0.979±0.017 | 0.028±0.022 | 31.64±2.87 | 0.948±0.038 | 0.080±0.050 | 98.93±21.68 |
| 3-9 | Ours | 3 | ✓ | ✓ | ✓ | 33.64±2.43 | 0.979±0.017 | 0.028±0.022 | 31.64±2.87 | 0.948±0.038 | 0.080±0.050 | 92.94±24.23 |
| 3-10 | Oracle | 3 | – | – | ✓ | 33.61±2.42 | 0.979±0.017 | 0.028±0.021 | 31.65±2.86 | 0.949±0.038 | 0.079±0.049 | 92.67±24.05 |

## D.1 ABLATIONS ON ODE NUMERICAL INTEGRATION

We study the effect of ODE numerical integration used when sampling from our generative model. Specifically, we ablate the algorithms (Euler and Heun), the step size (or equivalently the number of steps) used during the numerical integration, and the numerical precision of the model (float32 and bfloat16) during sampling. We provide quantitative results in Sec. S7. The results suggest our generative model is robust to numerical integration — we observe small change in performance when switching from the second-order method Heun with 100 steps using float32 (conditioning view FID = 6.6), to a relatively cheaper first-order Euler with 25 steps using bfloat16 (conditioning view FID = 6.7).

Table S4: **Reconstruction on PBR-Objaverse.** For 3D assets, we adapt inputs per model. TREL-LIS (Xiang et al., 2025) takes the ground-truth mesh and 150 sphere-distributed renderings. Ours uses RGBD images from 150 evenly distributed views. For appearance evaluation, we render each model's output from 100 random cameras, varying difficulty by adjusting camera radius. Each model is further evaluated under three distinct lighting conditions. Importantly, no separate models are trained; all evaluations are conducted on the same model. As a result, we conduct evaluations at the scale of 200 (objects) $\times$ 100 (views) $\times$ 2 (difficulties) $\times$ 3 (lightings) $\approx$ **120 thousand images**. We report Chamfer distances multiplied by $10^4$ for readability, computed using 100k sampled points each from the ground truth and reconstruction. We report in the format of mean±std, where the standard deviation is computed across objects.

| | Method | SH Deg | Enc Ray | Pred Occ | Mesh | Simple, Camera Radius [3, 4] | | | Hard, Camera Radius [1, 3] | | | CD (100k)↓ |
| | | | | | | PSNR↑ | SSIM↑ | LPIPS↓ | PSNR↑ | SSIM↑ | LPIPS↓ | |
|---|---|---|---|---|---|---|---|---|---|---|---|---|
| | | | | | | **Uniform Lighting** | | | | | | |
| 1-1 | TRELLIS | 0 | ✗ | - | ✓ | 28.63±3.09 | 0.955±0.028 | 0.046±0.025 | 25.06±2.93 | 0.902±0.057 | 0.121±0.062 | 98.09±22.21 |
| 1-2 | Ours | 0 | ✗ | ✗ | ✗ | 32.95±2.87 | 0.974±0.018 | 0.033±0.020 | 30.07±3.02 | 0.939±0.042 | 0.087±0.051 | 95.08±22.61 |
| 1-3 | Ours | 0 | ✓ | ✗ | ✗ | 33.14±2.68 | 0.974±0.018 | 0.034±0.021 | 30.21±2.85 | 0.937±0.042 | 0.089±0.053 | 94.16±22.55 |
| 1-4 | Ours | 1 | ✓ | ✗ | ✗ | 34.35±2.37 | 0.978±0.016 | 0.028±0.018 | 31.67±2.67 | 0.947±0.038 | 0.076±0.046 | 93.48±22.55 |
| 1-5 | Ours | 2 | ✓ | ✗ | ✗ | 34.47±2.45 | 0.978±0.016 | 0.028±0.018 | 31.74±2.73 | 0.947±0.039 | 0.077±0.047 | 94.01±22.69 |
| 1-6 | Ours | 3 | ✓ | ✗ | ✗ | 34.62±2.33 | 0.979±0.016 | 0.028±0.018 | 31.98±2.64 | 0.948±0.039 | 0.075±0.047 | 92.92±22.81 |
| 1-7 | Ours | 3 | ✗ | ✗ | ✗ | 34.66±2.62 | 0.979±0.016 | 0.029±0.018 | 31.83±2.89 | 0.948±0.039 | 0.077±0.047 | 94.71±22.50 |
| 1-8 | Ours | 3 | ✓ | ✓ | ✗ | 34.63±2.33 | 0.979±0.016 | 0.027±0.017 | 32.01±2.64 | 0.948±0.039 | 0.074±0.047 | 92.89±22.77 |
| 1-9 | Ours | 3 | ✓ | ✓ | ✓ | 34.63±2.33 | 0.979±0.016 | 0.027±0.017 | 32.01±2.64 | 0.948±0.039 | 0.075±0.047 | 85.82±25.06 |
| 1-10 | Oracle | 3 | – | – | ✓ | 34.64±2.31 | 0.979±0.016 | 0.027±0.017 | 32.07±2.62 | 0.949±0.038 | 0.074±0.046 | 85.57±24.80 |
| | | | | | | **TRELLIS Lighting** | | | | | | |
| 2-1 | TRELLIS | 0 | ✗ | - | ✓ | 29.69±2.59 | 0.958±0.025 | 0.044±0.023 | 26.03±2.50 | 0.904±0.053 | 0.118±0.058 | 95.16±20.77 |
| 2-2 | Ours | 0 | ✗ | ✗ | ✗ | 30.35±3.01 | 0.965±0.023 | 0.039±0.023 | 27.39±3.18 | 0.921±0.049 | 0.102±0.056 | 96.06±21.91 |
| 2-3 | Ours | 0 | ✓ | ✗ | ✗ | 30.37±3.04 | 0.965±0.023 | 0.040±0.023 | 27.41±3.21 | 0.919±0.050 | 0.102±0.056 | 95.11±21.89 |
| 2-4 | Ours | 1 | ✓ | ✗ | ✗ | 32.52±2.45 | 0.975±0.017 | 0.031±0.019 | 29.87±2.70 | 0.939±0.042 | 0.084±0.049 | 94.59±21.73 |
| 2-5 | Ours | 2 | ✓ | ✗ | ✗ | 32.47±2.45 | 0.975±0.018 | 0.031±0.019 | 29.90±2.73 | 0.940±0.042 | 0.083±0.049 | 94.98±21.91 |
| 2-6 | Ours | 3 | ✓ | ✗ | ✗ | 32.63±2.38 | 0.976±0.017 | 0.030±0.018 | 30.14±2.69 | 0.941±0.042 | 0.081±0.049 | 94.08±22.01 |
| 2-7 | Ours | 3 | ✗ | ✗ | ✗ | 32.56±2.72 | 0.975±0.018 | 0.031±0.019 | 29.89±2.97 | 0.939±0.042 | 0.084±0.049 | 95.38±21.90 |
| 2-8 | Ours | 3 | ✓ | ✓ | ✗ | 32.63±2.37 | 0.976±0.017 | 0.030±0.018 | 30.16±2.69 | 0.942±0.042 | 0.080±0.049 | 94.11±22.01 |
| 2-9 | Ours | 3 | ✓ | ✓ | ✓ | 32.62±2.37 | 0.976±0.017 | 0.030±0.018 | 30.16±2.69 | 0.942±0.042 | 0.080±0.049 | 87.17±24.29 |
| 2-10 | Oracle | 3 | – | – | ✓ | 32.61±2.37 | 0.976±0.017 | 0.029±0.018 | 30.20±2.69 | 0.942±0.042 | 0.080±0.048 | 87.02±24.19 |
| | | | | | | **Random Lighting** | | | | | | |
| 3-1 | TRELLIS | 0 | ✗ | - | ✓ | 26.29±3.56 | 0.939±0.038 | 0.052±0.030 | 22.74±3.37 | 0.869±0.075 | 0.134±0.070 | 99.60±24.34 |
| 3-2 | Ours | 0 | ✗ | ✗ | ✗ | 28.58±3.65 | 0.957±0.031 | 0.043±0.028 | 25.66±3.87 | 0.904±0.066 | 0.107±0.065 | 95.14±22.72 |
| 3-3 | Ours | 0 | ✓ | ✗ | ✗ | 28.88±3.61 | 0.956±0.032 | 0.043±0.028 | 25.93±3.81 | 0.903±0.067 | 0.109±0.066 | 94.53±22.73 |
| 3-4 | Ours | 1 | ✓ | ✗ | ✗ | 30.36±3.15 | 0.965±0.027 | 0.036±0.024 | 27.60±3.43 | 0.920±0.059 | 0.095±0.058 | 93.98±22.70 |
| 3-5 | Ours | 2 | ✓ | ✗ | ✗ | 30.39±3.08 | 0.965±0.027 | 0.036±0.024 | 27.65±3.39 | 0.920±0.060 | 0.095±0.059 | 94.72±22.86 |
| 3-6 | Ours | 3 | ✓ | ✗ | ✗ | 30.59±3.08 | 0.966±0.027 | 0.036±0.024 | 27.92±3.42 | 0.922±0.059 | 0.093±0.059 | 93.41±22.84 |
| 3-7 | Ours | 3 | ✗ | ✗ | ✗ | 30.11±3.48 | 0.964±0.027 | 0.037±0.024 | 27.27±3.75 | 0.917±0.060 | 0.096±0.059 | 94.74±22.64 |
| 3-8 | Ours | 3 | ✓ | ✓ | ✗ | 30.59±3.09 | 0.966±0.027 | 0.035±0.024 | 27.94±3.43 | 0.922±0.060 | 0.092±0.059 | 93.39±22.82 |
| 3-9 | Ours | 3 | ✓ | ✓ | ✓ | 30.60±3.08 | 0.966±0.027 | 0.035±0.024 | 27.95±3.43 | 0.922±0.059 | 0.092±0.059 | 85.73±24.79 |
| 3-10 | Oracle | 3 | – | – | ✓ | 30.59±3.07 | 0.966±0.026 | 0.035±0.024 | 27.97±3.42 | 0.922±0.059 | 0.092±0.058 | 85.56±24.65 |

## D.2 RUNTIME AND MEMORY ANALYSIS

We analyze the runtime for both TRELLIS and our generative models in Tab. S8. Our model's latent sampling costs 9.3 seconds on while all decoders' feedforward passes cost less than 100 milliseconds on a single NVIDIA H100 80GB HBM3 GPU. In comparison, for TRELLIS, sampling SLAT (both coarse voxel and feature) takes 11.8 seconds. Utilizing one-step flow-matching models like MeanFlow (Geng et al., 2025) can further improve the speed of our generative model and is left as future work.

Table S5: **Ablation on number of input views for reconstruction during inference.** We choose TRELLIS lighting setup on Toys4k dataset. Our model is the same as "ours" in Tab. 1. Both TRELLIS and ours are trained with 150 views. For appearance evaluation, we render each model's output from 100 random cameras, varying difficulty by adjusting camera radius. We report in the format of mean$\pm$std, where the standard deviation is computed across objects. We report Chamfer distances multiplied by $10^4$ for readability, computed using 100k sampled points each from the ground truth and reconstruction. Note, we re-render the evaluation data for this ablation, thus row 1 (row 2) differs slightly from row 2-1 (row 2-9) in Tab. S2.

| | Method | Simple, Camera Radius [3, 4] | | | Hard, Camera Radius [1, 3] | | | CD (100k)↓ |
|---|---|---|---|---|---|---|---|---|
| | | PSNR↑ | SSIM↑ | LPIPS↓ | PSNR↑ | SSIM↑ | LPIPS↓ | |
| | | **150 input views** | | | | | | |
| 1 | TRELLIS | 31.559$\pm$3.509 | 0.9740$\pm$0.0224 | 0.0361$\pm$0.0217 | 27.948$\pm$3.539 | 0.9408$\pm$0.0508 | 0.0928$\pm$0.0539 | 90.65$\pm$25.13 |
| 2 | Ours | 33.909$\pm$3.157 | 0.9841$\pm$0.0162 | 0.0260$\pm$0.0189 | 32.073$\pm$3.521 | 0.9658$\pm$0.0403 | 0.0585$\pm$0.0458 | 80.54$\pm$27.62 |
| | | **120 input views** | | | | | | |
| 3 | TRELLIS | 31.518$\pm$3.509 | 0.9738$\pm$0.0225 | 0.0363$\pm$0.0218 | 27.912$\pm$3.541 | 0.9404$\pm$0.0510 | 0.0932$\pm$0.0541 | 90.88$\pm$25.19 |
| 4 | Ours | 33.908$\pm$3.158 | 0.9841$\pm$0.0162 | 0.0260$\pm$0.0188 | 32.072$\pm$3.522 | 0.9658$\pm$0.0403 | 0.0585$\pm$0.0457 | 80.53$\pm$27.58 |
| | | **90 input views** | | | | | | |
| 5 | TRELLIS | 31.431$\pm$3.506 | 0.9734$\pm$0.0227 | 0.0366$\pm$0.0221 | 27.833$\pm$3.540 | 0.9397$\pm$0.0514 | 0.0938$\pm$0.0545 | 91.19$\pm$25.16 |
| 6 | Ours | 33.910$\pm$3.157 | 0.9841$\pm$0.0162 | 0.0260$\pm$0.0189 | 32.074$\pm$3.522 | 0.9658$\pm$0.0403 | 0.0585$\pm$0.0457 | 80.53$\pm$27.60 |
| | | **60 input views** | | | | | | |
| 7 | TRELLIS | 31.270$\pm$3.496 | 0.9726$\pm$0.0231 | 0.0372$\pm$0.0224 | 27.688$\pm$3.533 | 0.9383$\pm$0.0520 | 0.0952$\pm$0.0552 | 91.92$\pm$25.16 |
| 8 | Ours | 33.909$\pm$3.155 | 0.9841$\pm$0.0162 | 0.0260$\pm$0.0188 | 32.073$\pm$3.519 | 0.9658$\pm$0.0403 | 0.0585$\pm$0.0457 | 80.53$\pm$27.60 |
| | | **30 input views** | | | | | | |
| 9 | TRELLIS | 30.692$\pm$3.441 | 0.9699$\pm$0.0244 | 0.0396$\pm$0.0238 | 27.159$\pm$3.484 | 0.9336$\pm$0.0541 | 0.1002$\pm$0.0576 | 94.58$\pm$25.50 |
| 10 | Ours | 33.908$\pm$3.157 | 0.9841$\pm$0.0162 | 0.0260$\pm$0.0188 | 32.072$\pm$3.521 | 0.9658$\pm$0.0403 | 0.0585$\pm$0.0457 | 80.56$\pm$27.61 |

Table S6: **Single-image-conditioned generation on Toys4k with TRELLIS lighting.** KID is reported by $\times 100$. CFG scale is 3.0. The best is highlighted.

| | Method | Train w/ Ray | Infer w/ GT Ray | Train Iters | CLIP↑ | Conditioning View | | | | Novel View | | | |
|---|---|---|---|---|---|---|---|---|---|---|---|---|---|
| | | | | | | FID↓ | KID↓ | FID$_{dino}$↓ | KID$_{dino}$↓ | FID↓ | KID↓ | FID$_{dino}$↓ | KID$_{dino}$↓ |
| 1 | TRELLIS | ✗ | - | 400k | 0.899$\pm$0.045 | 12.84 | 0.088 | 84.692 | 2.311 | 7.600 | 0.100 | 67.458 | 3.166 |
| 2-1 | Ours | ✗ | - | 280k | 0.906$\pm$0.040 | 8.193 | 0.012 | 48.117 | 0.461 | 6.648 | 0.064 | 75.814 | 4.321 |
| 2-2 | Ours | ✗ | - | 400k | 0.906$\pm$0.041 | 7.741 | 0.010 | 44.555 | 0.392 | 6.413 | 0.064 | 71.436 | 3.997 |
| 2-3 | Ours | ✗ | - | 600k | 0.905$\pm$0.041 | 6.219 | 0.009 | 41.621 | 1.333 | 6.216 | 0.058 | 66.530 | 3.522 |
| 3 | Ours | ✓ | ✗ | 290k | 0.900$\pm$0.040 | 10.78 | 0.066 | 65.644 | 2.281 | 8.076 | 0.101 | 92.915 | 6.698 |
| 4 | Ours | ✓ | ✓ | 290k | 0.904$\pm$0.039 | 10.13 | 0.053 | 61.342 | 1.665 | 7.831 | 0.097 | 86.091 | 5.826 |

Table S7: **Ablation on DiT sampler for single-image-conditioned generation.** The experiments are conducted on Toys4k with TRELLIS lighting. The generative model is trained for 600k iterations. Note, row 1 is copied from "ours" in Tab. 3 . KID is reported by $\times 100$. CFG scale is 3.0. Our generative model's performance is robust across various numbers of sampling steps and numerical integration algorithms.

| | Occ Pred | Data Type | Method | Step | CLIP↑ | Conditioning View | | | | Novel View | | | |
|---|---|---|---|---|---|---|---|---|---|---|---|---|---|
| | | | | | | FID↓ | KID↓ | FID$_{dino}$↓ | KID$_{dino}$↓ | FID↓ | KID↓ | FID$_{dino}$↓ | KID$_{dino}$↓ |
| 1 | ✗ | float32 | Heun | 100 | 0.905$\pm$0.041 | 6.219 | 0.009 | 41.621 | 1.333 | 6.216 | 0.058 | 66.530 | 3.522 |
| 2 | ✓ | float32 | Heun | 100 | 0.905$\pm$0.041 | 6.622 | 0.021 | 42.197 | 1.391 | 6.270 | 0.064 | 66.699 | 3.534 |
| 3 | ✓ | bfloat16 | Heun | 100 | 0.905$\pm$0.041 | 6.661 | 0.020 | 43.992 | 1.741 | 6.270 | 0.063 | 68.025 | 3.906 |
| 4 | ✓ | bfloat16 | Heun | 50 | 0.905$\pm$0.041 | 6.659 | 0.020 | 45.533 | 2.105 | 6.266 | 0.062 | 68.319 | 4.185 |
| 5 | ✓ | bfloat16 | Heun | 25 | 0.904$\pm$0.041 | 6.644 | 0.019 | 54.231 | 4.011 | 6.251 | 0.060 | 77.148 | 5.879 |
| 6 | ✓ | bfloat16 | Euler | 100 | 0.906$\pm$0.041 | 6.656 | 0.022 | 42.472 | 1.476 | 6.365 | 0.066 | 67.856 | 3.848 |
| 7 | ✓ | bfloat16 | Euler | 50 | 0.905$\pm$0.041 | 6.688 | 0.023 | 42.363 | 1.430 | 6.384 | 0.066 | 68.987 | 3.958 |
| 8 | ✓ | bfloat16 | Euler | 25 | 0.905$\pm$0.041 | 6.733 | 0.025 | 43.034 | 1.280 | 6.833 | 0.074 | 75.687 | 4.484 |

Table S8: **Generative model runtime analysis.** All results are reported with `torch.profiler` across three runs. TRELLIS uses 50 Euler steps for both its sparse structure and structured latent generations. We use 50 Euler steps for generating the latents, corresponding to row 7 in Tab. S7.

| | Cond Proc (ms) | Structure Gen (s) | Latent Gen (s) | Occ Pred (ms) | 3DGS Dec (ms) | Mesh Dec (ms) | Total (s) | Memory (GB) |
|---|---|---|---|---|---|---|---|---|
| **NVIDIA A100-SXM4-80GB** | | | | | | | | |
| TRELLIS | 68.90±0.49 | 4.89±0.80 | 7.720±5.10 | – | 18.70±5.46 | 67.33±13.98 | 12.76 | 12.70 |
| Ours | 68.78±0.41 | – | 17.32±1.50 | 36.07±3.67 | 35.32±14.2 | 90.78±29.75 | 17.55 | 15.95 |
| **NVIDIA H100 80GB HBM3** | | | | | | | | |
| TRELLIS | 31.01±0.55 | 3.95±1.17 | 7.868±4.06 | – | 15.03±6.19 | 46.81±13.31 | 11.91 | 12.69 |
| Ours | 22.58±10.3 | – | 9.266±0.38 | 27.16±6.87 | 30.96±14.3 | 79.15±31.71 | 9.426 | 15.93 |

# E  IMPLEMENTATION DETAILS

## E.1  ARCHITECTURES

We provide detailed network architectures in Fig. S2 to S7. These include our encoder (Sec. 3.3) in Fig. S2, velocity decoder and Gaussian decoder (Sec. 3.4) in Fig. S3 and S4, occupancy decoder in Fig. S5, mesh decoder in Fig. S6, and generative model's DiT (Sec. 3.5) in Fig. S7.

## E.2  POSITION ENCODING

We have the following position encoding function applied on *each channel* of the input data:

$$\{\sin(u_0), \ldots, \sin(u_{F-1}), \cos(u_0), \ldots, \cos(u_{F-1})\}, \tag{S2}$$

$$\text{where } u_i = x \cdot 2^{\left(M_{\min} + i \cdot \frac{M_{\max} - M_{\min}}{F-1}\right)}, \tag{S3}$$

$x$ is the value at the corresponding channel where the position encoding is applied. We use $F = 32$, $M_{\min} = 0$, $M_{\max} = 12, 8$, and 8 in position encoding functions for 3D location $\mathbf{x}_i$, viewing direction $\hat{\mathbf{d}}_i$, color $\mathbf{c}_i$ in Eq. (1) respectively. For time step $t$ in flow matching (Eq. (2)), we use $F = 16$, $M_{\min} = \log_2 2\pi$, and $M_{\max} = M_{\min} + F - 1$.

## E.3  3D GAUSSIAN PREDICTION

In Fig. S4, the output position of 3D Gaussian is predicted with respect to a normalized space centered around the occupied voxel's world coordinates, and is then translated to the world coordinate system using the voxel's information. Specifically, we predict 3D Gaussian's position as $\mathbf{x}_{\text{output}} \in [-1, 1]^3$. Assume the corresponding voxel's center is located at $\mathbf{x}_{\text{voxel}} \in \mathbb{R}^3$ in the world coordinate system. The final 3D Gaussian's position in the world coordinate system is computed as $\mathbf{x}_{\text{3DGS}} = \mathbf{x}_{\text{voxel}} + s \cdot \mathbf{x}_{\text{output}}$, where $s$ is a hyperparameter to define the size of the normalized space mentioned above. In our experiments, we set $s = 0.05$. Note, $s = 0.05$ is actually larger than the voxel size we consider. This is intentional as it provides more flexibility, such that the predicted 3D Gaussian can go across the voxel boundaries.

## E.4  TOKENIZER TRAINING

Our tokenizer is trained with the following loss:

$$\mathcal{L}_{\text{tokenizer}} = \mathcal{L}_{\text{geo}}(\boldsymbol{\theta}) + \mathcal{L}_{\text{radiance}}(\boldsymbol{\theta}) + 10^{-4} \cdot \text{KL}\left(q(\mathcal{S}|\mathcal{X})|p(\mathcal{S})\right), \tag{S4}$$

where $\mathcal{L}_{\text{geo}}(\boldsymbol{\theta})$ and $\mathcal{L}_{\text{radiance}}(\boldsymbol{\theta})$ are from Eq. (2) and Eq. (3), respectively.

We use the KL-divergence $\text{KL}\left(q(\mathcal{S}|\mathcal{X})|p(\mathcal{S})\right)$ to regularize the latent space, where $p(\mathcal{S})$ and $q(\mathcal{S}|\mathcal{X})$ represent prior and posterior distribution for the latent representation $\mathcal{S}$. Ideally, this should be imposed on the joint distribution of the $k$ latent vectors (Sec. 3.2). In practice, we simplify and assume each element in the latent space is independent. Thus we have

$$\text{KL}\left(q(\mathcal{S}|\mathcal{X})|p(\mathcal{S})\right) = \sum_{i=1}^{k=8192} \sum_{j=1}^{d=32} \text{KL}\left(q(s_{i,j}|\mathcal{X})|q(s_{i,j})\right), \tag{S5}$$

where $s_{i,j}$ is the $j$-th element of the $i$-th latent vector $\mathbf{s}_i \in \mathcal{S}$. Further, we assume $p(s_{i,j})$ follows $\mathcal{N}(0,1)$ while $q(s_{i,j}|\mathcal{X})$ is $\mathcal{N}(s_{i,j}, 10^{-6})$. Thus, KL $\left(q(s_{i,j}|\mathcal{X})|q(s_{i,j})\right)$ boils down to $\|s_{i,j}\|^2$.

In practice, we find that Eq. (S5) is effective. When computing the mean and standard deviation of the latent representations $\mathcal{S}$ across 1000 objects, we obtain values of 1.1300 and 19.8604. In contrast, with the proposed regularization, the latent space becomes more compact: the mean and standard deviation decrease to 0.0931 and 1.7800, without compromising reconstruction performance.

### E.5 OCCUPANCY DECODER TRAINING

We adopt the pretrained sparse-structure VAE from TRELLIS (Xiang et al., 2025) to compute the occupancy grid. Specifically, given a LiTo latent code, we first generate a low-resolution continuous latent representation (in our case, $16^3$). We then leverage the TRELLIS decoder to upsample this representation and predict occupancy values on a higher-resolution grid (in our case, $64^3$). Our model is specified in Fig. S5. The details of the upsampling decoder is specified in "3D convolutional U-net" in Sec. A.1 from Xiang et al. (2025).

The training loss is defined as per-element Huber loss between 1) our model predicted low-resolution representation, and 2) the representation encoded from the ground-truth occupancy grid with the pretrained encoder from TRELLIS.

### E.6 MESH DECODER TRAINING

We train the mesh decoder in Fig. S6 primarily by penalizing the discrepancy between renderings generated from the ground-truth mesh and those from the estimated mesh. For each mesh encountered during training, we randomly sample 12 views for supervision. Specifically, we use the following loss:

$$\mathcal{L}_{\text{mesh}} = \mathcal{L}_{\text{mask}} + 10 \cdot \mathcal{L}_{\mathbf{x}_{\text{cam}}} + \mathcal{L}_{\text{face\_normal}} + \mathcal{L}_{\text{vertex\_normal}} + \mathcal{L}_{\text{reg\_dev}} + \mathcal{L}_{\text{reg\_sdf}}. \tag{S6}$$

$\mathcal{L}_{\text{mask}}$ is the $\ell_1$ distance between 1) mask rendered from the ground-truh mesh; and 2) mask rendered from the generated mesh.

$\mathcal{L}_{\mathbf{x}_{\text{cam}}}$ is for each pixel's corresponding XYZ in the camera coordinate system. Concretely, we compute the $\ell_2$ distance between the ground-truth coordinates and the predicted ones. We only apply this loss to pixels within the ground-truth object mask area. Compared to using depth loss alone, this provides a stronger supervisory signal, which we found significantly improves training performance.

$\mathcal{L}_{\text{face\_normal}}$ and $\mathcal{L}_{\text{vertex\_normal}}$ supervise the predicted face normals and vertex normals, respectively. They are computed as 1 minus the cosine similarity (*i.e.* negative cosine distance) between the predicted normal maps and the corresponding ground-truth normal maps. Similar to $\mathcal{L}_{\mathbf{x}_{\text{cam}}}$, we only apply this loss to pixels within the ground-truth object mask area.

Since we use FlexiCubes (Shen et al., 2023) to produce the mesh, we inherit its regularizations during training. $\mathcal{L}_{\text{reg\_dev}}$ penalizes deviations between each dual vertex and the edge crossings that define the face containing it (see Eq. (8) in (Shen et al., 2023)).[3] $\mathcal{L}_{\text{reg\_sdf}}$ penalizes sign changes of the SDF across all grid edges. [4]

### E.7 TRAINING SETUPS

We use AdamW (Loshchilov & Hutter, 2019) for all our training.

For our tokenizer training, *i.e.*, Eq. (S4), that involves encoder (Fig. S2), velocity decoder (Fig. S3), and 3D Gaussians decoder (Fig. S4), we use $\beta_1 = 0.9$, and $\beta_2 = 0.98$ without weight decay for AdamW. We use the following learning rate scheduler copied from Sec. 5.3 in Vaswani et al. (2017):

$$\text{lr}(\text{step}) = 0.4 \cdot d_{\text{model}}^{-0.5} \cdot \min\left(\text{step}^{-0.5}, \frac{\text{step}}{\text{warmup\_steps}^{1.5}}\right), \tag{S7}$$

---

[3] https://github.com/nv-tlabs/FlexiCubes/blob/4cc7d6c3d0ce/examples/optimize.py#L112-L113
[4] https://github.com/nv-tlabs/FlexiCubes/blob/4cc7d6c3d0ce/examples/loss.py#L96

where $d_{\text{model}} = 512$, which is our latent dimension for Perceiver IO, and warmup_steps $= 4000$ in our case. The model is trained with an effective batch size of 256 for 90k iterations on 64 H100 GPUs for 9 days.

For the occupancy decoder training that involves Fig. S5, we use the same optimizer and learning rate setup as tokenzer training. We train the model for 58k iterations with an effective batch size 256 on 64 H100 GPUs for 1.5 days.

For the mesh decoder training, *i.e.*, Eq. (S6), that involves Fig. S6, we use the same optimizer and learning rate setup as tokenizer training. We train the model for 100k iterations with an effective batch size 128 on 64 H100 for 3 days.

For the generative model training that involves Fig. S7, we use $\beta_1 = 0.9$, and $\beta_2 = 0.999$, and a weight decay of 0.01 for AdamW. We use the following linear warmup scheduler for the learning rate:

$$\text{lr(step)} = \begin{cases} 10^{-6} + \frac{\text{step}}{\text{warmup\_steps}} \cdot (10^{-4} - 10^{-6}) & \text{if step} \leq \text{warmup\_steps} \\ 10^{-4} & \text{otherwise} \end{cases}, \tag{S8}$$

where we have warmup_steps $= 5000$ in our case. We train the model for 600k iterations with an effective batch size 256 on 128 H100 GPUs for 20 days.

## F  MORE STUDIES

### F.1  STUDYING SPHERICAL HARMONICS DEGREES

Our Gaussian decoder outputs Gaussians with spherical harmonics up to degree three. We study what information is captured by individual spherical harmonics degrees. In Fig. S8 and Fig. S9, we render the 3D Gaussians from both reconstruction and generation by clipping the degree of the spherical harmonics (*i.e.*, we use only $\leq 3$ degrees during rendering). We observe that zeroth-degree renderings are mostly view-independent and have little lighting baked in, whereas higher-degree renderings illustrate lighting effects. This is in contrast to TRELLIS's results whose zeroth-degree renderings contain both baked lighting and inaccurate view-dependent appearance produced using micro-surface geometry (Walter et al., 2007). The results suggest that our model is able to represent view-dependent effects using the higher-degree spherical harmonics, and to use the zeroth-degree rendering for view-independent, diffused, appearance. This separation is an interesting finding, and it provides potential opportunity for future investigation of relighting using our representation.

## G  AUTHOR CONTRIBUTIONS

All authors contributed to writing this paper, designing the experiments, and discussing results at each stage of the project.

**Framing.** Oncel Tuzel led the research direction, including research framing and question identification. All authors contributed to setting project priorities.

**Writing.** Jen-Hao Rick Chang and Xiaoming Zhao completed the majority of the writing while Dorian Chan refined and polished the presentation.

**Data.** Jen-Hao Rick Chang and Xiaoming Zhao developed the data rendering scripts to convert 3D assets into the surface light fields required for Eq. (1). Xiaoming Zhao developed the data preprocessing pipeline to clean and curate the dataset to ensure high quality inputs for model training. Jen-Hao Rick Chang implemented the efficient dataloader based on `webdataset` (webdataset development team, 2026).

**Model design.** Jen-Hao Rick Chang and Xiaoming Zhao were primarily responsible for developing the encoder (Sec. 3.3), decoder (Sec. 3.4), and image-to-3D generative model (Sec. 3.5). Jen-Hao Rick Chang designed and implemented the data structures that enable the efficient 3D patchification process in Fig. 3, designed encoder, velocity decoder, and iterates on Gaussian decoders, and he built the 3D library, `plibs`.

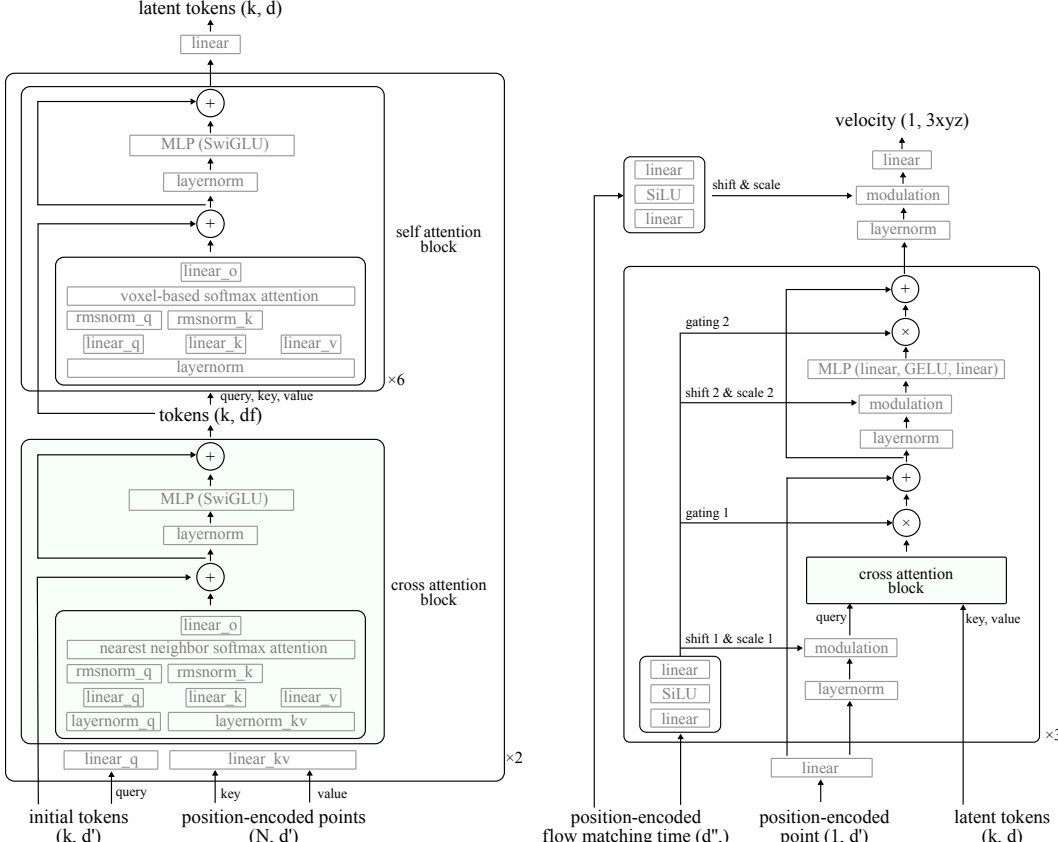

Figure S2: **Encoder architecture.** The model uses a feature dimension of $df = 512$, while the hidden layer in MLP uses a feature dimension of 2048. The number of heads for cross-attention and self-attention is 16. The input dimension $d' = 396$, which includes 3D location, position-encoded 3D location, RGB, position-encoded RGB, and Plucker coordinates. Our latent has $k = 8192$ and $d = 32$. Please refer to Sec. E for position encoding details.

Figure S3: **Velocity decoder architecture.** The model uses a feature dimension of 512, while the hidden layer in MLP uses a feature dimension of 2048. The number of heads for cross-attention is 8. Our latent has $k = 8192$ and $d = 32$. We have $d' = 195$, which includes 3D location and position-encoded 3D location. Meanwhile, $d'' = 64$, which is obtained by applying a linear layer to time-step position encoding in Eq. (S2). Please refer to Sec. E for position encoding details.

**Model training.** Jen-Hao Rick Chang and Xiaoming Zhao led the training of all models described in this study, including both tokenizers and the generative models presented in the paper. Specifically, Jen-Hao Rick Chang trained the tokenizer (Fig. S2, Fig. S3, and Fig. S4), occupancy prediciton decoder (Fig. S5), and generative model (Fig. S7). Xiaoming Zhao trained the tokenizer (Fig. S2, Fig. S3, and Fig. S4), mesh decoder (Fig. S6), and generative model (Fig. S7). Jen-Hao Rick Chang implemented the training backbone.

**Evaluation.** Jen-Hao Rick Chang and Xiaoming Zhao led the evaluation strategy and experimental design. Xiaoming Zhao developed the evaluation pipeline and frameworks used to generate all quantitative results reported in the paper. Dorian Chan conducted the geometric evaluations for the baseline methods presented Tab. 2 while Xiaoming Zhao completed the remaining tables.

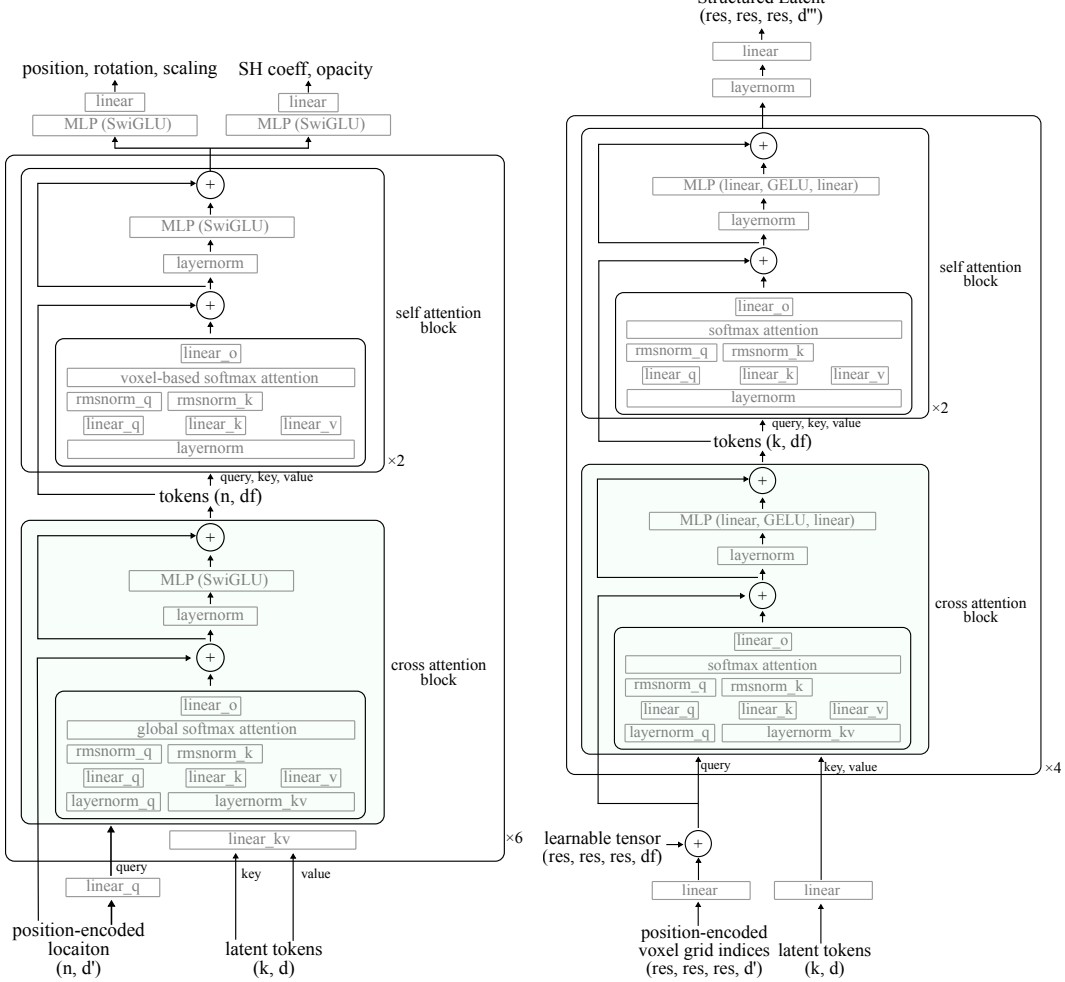

Figure S4: **3D Gaussian decoder architecture.** The model uses a feature dimension of $df = 512$, while the hidden layer in MLP uses a feature dimension of 2048. The number of heads for cross-attention and self-attention is 8. Our latent has $k = 8192$ and $d = 32$. We have $d' = 195$, which includes 3D location and position-encoded 3D location. Please refer to Sec. E for position encoding details.

Figure S5: **Occupancy decoder architecture.** The model uses a feature dimension of $df = 512$, while the feature dimension for QKV in cross/self-attention is 1024. The hidden layer in MLP uses a feature dimension of 2048. Our latent has $k = 8192$ and $d = 32$. The number of heads for cross-attention and self-attention is 8. We have $d' = 771$, with $M_{\min} = 0$, $M_{\max} = 5$, and $F = 128$ in Eq. (S2) for encoding 3D location. We use resolution of 16, *i.e.*, res $= 16$. Please refer to Sec. E for position encoding details.

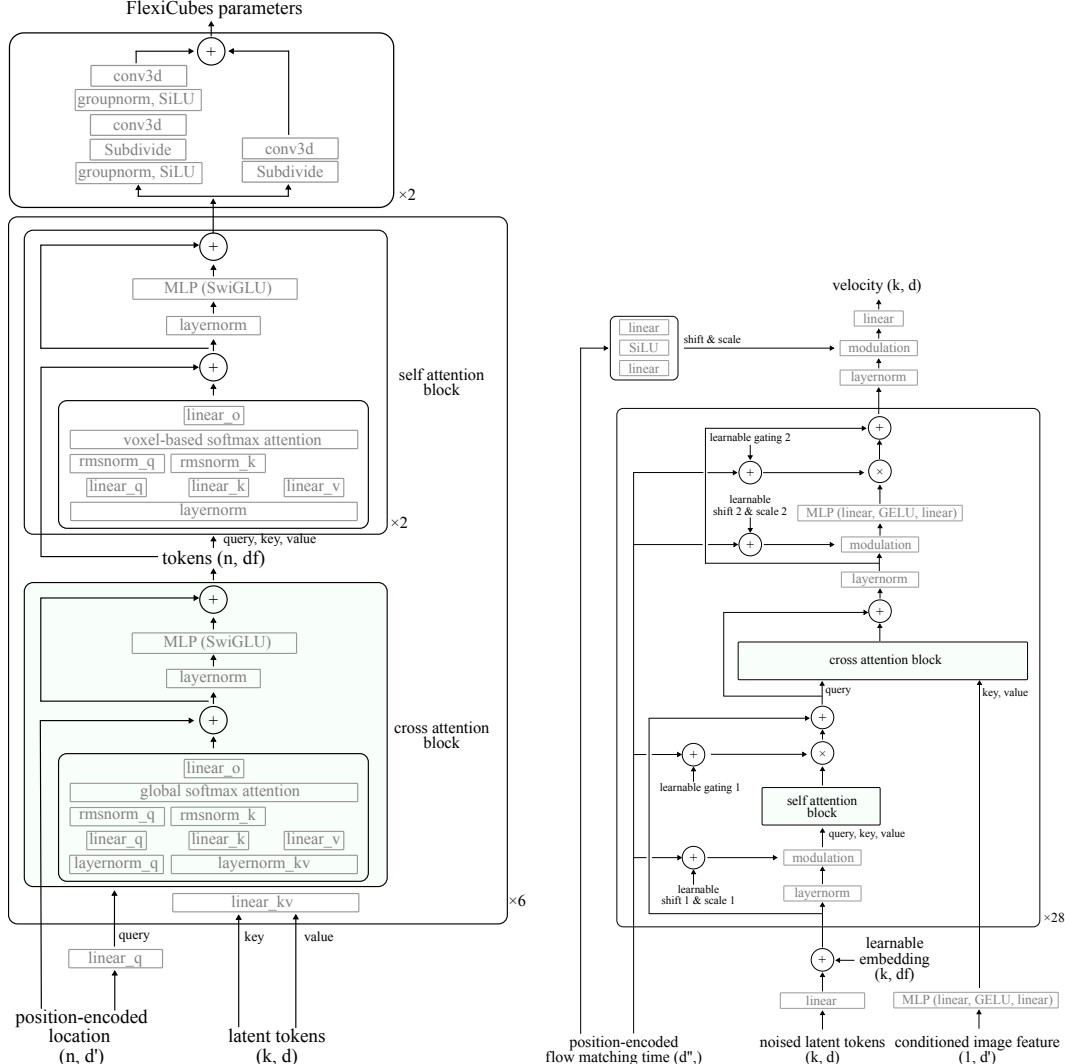

Figure S6: **Mesh decoder architecture.** The model uses a feature dimension of $df = 512$, while the hidden layer in MLP uses a feature dimension of 2048. Our latent has $k = 8192$ and $d = 32$. The number of heads for cross-attention and self-attention is 16. We have $d' = 195$, which includes 3D location and position-encoded 3D location. Please refer to Sec. E for position encoding details.

Figure S7: **Generative model DiT architecture.** The model uses a feature dimension of $df = 1152$, while the hidden layer in MLP uses a feature dimension of 4608. Our latent has $k = 8192$ and $d = 32$. The number of heads for self-attention and cross-attention is 16. The feature dimension for conditioning image $d' = 2048$. We have $d'' = 64$, with $M_{\min} = 0$, $M_{\max} = 12$, and $F = 32$ in Eq. (S2) for encoding flow matching time step. Please refer to Sec. E for position encoding details.

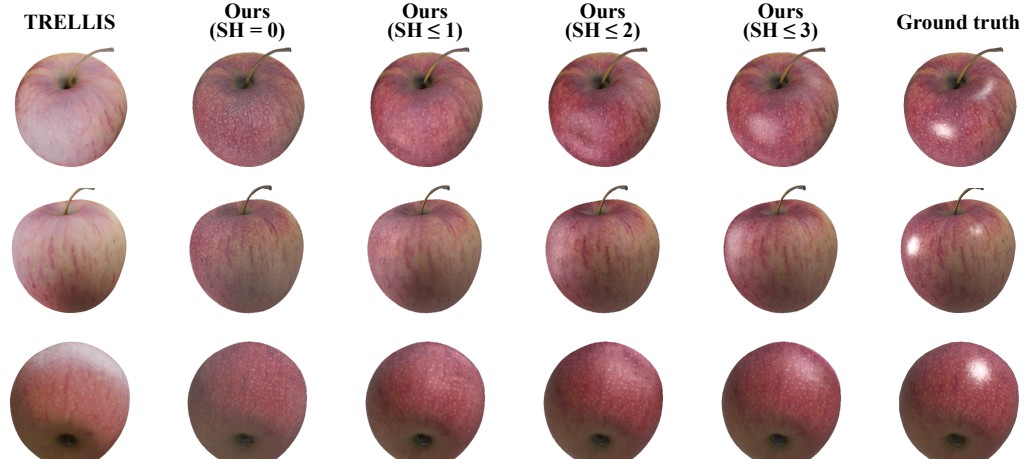

Figure S8: **Rendering with various spherical harmonics degrees in reconstruction.** When restricted to zeroth-order spherical harmonics, our 3D Gaussians produce a view-independent appearance and avoid the over-exposed regions observed in TRELLIS's renderings. As we progressively incorporate higher-order spherical harmonics, our method yields increasingly pronounced view-dependent effects. Mesh credit: DigitalSouls (2019).

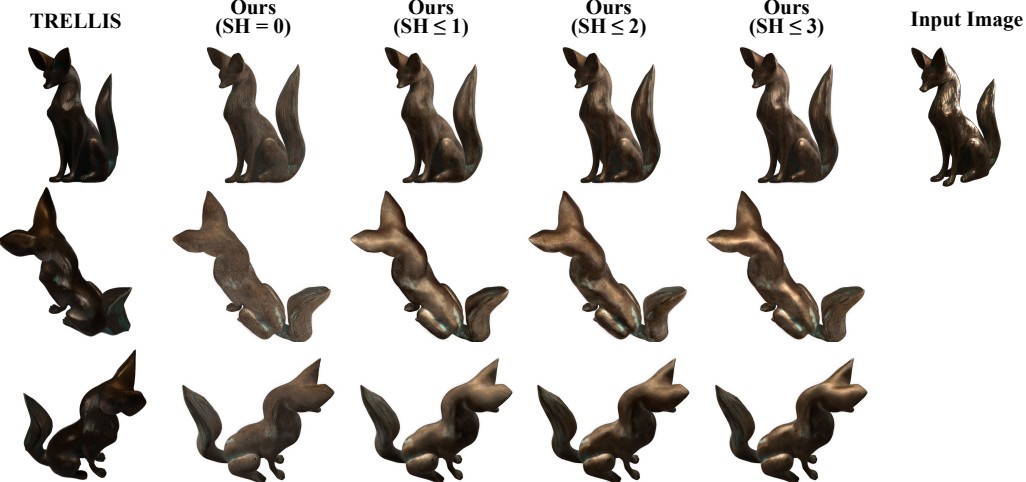

Figure S9: **Rendering with various spherical harmonics degrees in generation.** When restricted to zeroth-order spherical harmonics, our 3D Gaussians produce a view-independent appearance and avoid the over-exposed regions observed in TRELLIS's renderings. As we progressively incorporate higher-order spherical harmonics, our method yields increasingly pronounced view-dependent effects. Mesh credit: Eleanie (2025).

