# OpenReview forum: "LiTo: Surface Light Field Tokenization"
_ICLR.cc/2026/Conference — ICLR 2026 Poster_

### Official Review · Reviewer_wBM6 · 2025-10-27

**Soundness:** 3
**Presentation:** 3
**Contribution:** 2
**Rating:** 6
**Confidence:** 4

**Summary:**

This paper presents a 3D latent representation that simultaneously models object geometry and view-dependent appearance. Unlike previous methods that separately reconstruct geometry or predict view-independent appearance, this approach encodes RGB-depth samples of a surface light field into compact latent vectors. The unified latent space effectively captures realistic visual phenomena, including specular highlights and Fresnel reflections, under complex lighting conditions.
The main contribution lies in introducing a unified 3D latent representation that jointly learns geometry and view-dependent appearance from surface light field samples. Additionally, the authors propose a latent flow matching model conditioned on a single image, enabling the generation of 3D objects with lighting- and material-consistent appearances. Experimental results demonstrate superior visual realism and fidelity compared to existing approaches.

**Strengths:**

Originality: It is the first work to unify 3D geometry and view-dependent appearance in a single framework by encoding surface light fields sampled from multi-view RGB-D images. It employs 3D Gaussians with 3rd-order spherical harmonics (SH) to reproduce view-dependent effects such as specular highlights and Fresnel reflections, which traditional methods fail to capture. Additionally, it introduces a k-NN-based 3D patchification and voxel-aware attention structure, overcoming the efficiency bottleneck in processing million-scale surface light field samples.

Quality: The experimental design is rigorous and validated on datasets including ObjaverseXL and Toys4k. In reconstruction tasks, it achieves a PSNR of 36.45±4.00 (surpassing TRELLIS by 3.27), while in generation tasks, it reaches an FID of 8.193 at the conditioning view (better than TRELLIS’s 12.84). The dual-decoder supervision (flow-matching geometry decoder and 3D Gaussian appearance decoder) balances geometric accuracy and appearance realism, producing reproducible results.

Clarity: The framework follows a clear sampling–encoding–decoding–generation pipeline. Core formulas (e.g., the flow-matching loss and the radiance loss) are clearly annotated, and key concepts (e.g., surface light field, 3D Gaussians) are explained consistently. The connections between symbol definitions, method descriptions, and experimental procedures are well aligned, ensuring easy comprehension.

Significance: It fills the gap of “separated geometry and appearance modeling” in 3D latent representations, providing an efficient framework for 3D content generation and virtual-real fusion. Its ability to avoid precomputed coarse geometry reduces data dependency, and the view alignment of single-image-to-3D generation promotes the application of downstream tasks (e.g., game asset creation, AR modeling).

**Weaknesses:**

-1. In the expression $I_{gt}=Render(scene,H,E)$ presented below Eq. 3, the term scene is used, but its definition or corresponding description is not provided. Could you clarify what it represents?

-2. The results presented in Table 2 indicate that LiTo still exhibits a performance gap compared to TripoSF and TRELLIS. I understand that this may be due to the differences in the types of input data. The authors could frame this as a deliberate design choice, prioritizing appearance over geometry; however, this trade-off is a key characteristic of the method that should be more clearly articulated.

**Questions:**

-1. In this work, the authors utilize an RGB-D dataset. I wonder whether using an RGB-only dataset augmented with depth predicted by a large model could achieve reconstruction results comparable to those obtained with an RGB-D dataset. In other words, how significant is the impact of accurate depth information on the reconstruction quality?

-2. The manuscript mentions that LiTo uses 150 views during training. It would be informative if the authors could provide a brief discussion or evaluation of the reconstruction results when using fewer input views. Such an analysis could help assess the robustness of the method to reduced view availability.

-3. The paper appears to address two aspects: jointly modeling object geometry and view-dependent appearance, and single-view 3D reconstruction. These seem to correspond to two distinct tasks. Could the authors clarify which of these is the primary focus of the work? My understanding is that single-view 3D reconstruction is more akin to a downstream application of the learned model, so a clear explanation would help resolve this point of confusion.

---

> ### Author Response · Authors · 2025-11-22
>
> ## Writing clarification
>
> Thank you for noticing the issue. “scene” in L206 refers to the ground-truth object (i.e., the mesh). We have updated the PDF to make it clearer.
>
> ## About TRELLIS and TripoSF
>
> TRELLIS and TripoSF’s representations require coarse voxel geometry to be known.  When we compare with the their methods, as noted in Tab. 2, we provide ground-truth voxel geometry to these methods: TRELLIS expects $64^3$  and TripoSF $256^3$ occupancy grids as inputs.
>
> In comparison, our method does not utilize any ground-truth geometry information and infers both geometry and appearance.  Thus, it is not exactly an apple-to-apple comparison between ours and the two methods (and hence they are separated in two groups in Tab. 2).
>
> In the updated PDF, we have included a new experiment in which we decode meshes from our representation using a mesh decoder similar to TRELLIS (row 7-2 in Tab. 2). We evaluated the reconstructed mesh and showed that it outperforms both TRELLIS and TripoSF without taking any ground truth coarse geometry as input.
>
> ## About geometry performance
>
> We identified a problem in our previous evaluation set that resulted in input data issues, primarily producing a mismatch between the input surface points used by LiTo and the points used for other methods and for ground truth, while also creating normalization issues for a small set of objects. After correcting this bug, the LiTo numbers for geometry changed more noticeably, while the baseline methods also changed slightly but maintained essentially the same relative performance.
>
> Furthermore, in the original submission, we evaluated LiTo’s geometry only using points sampled from the flow-matching velocity decoder. To compare with existing techniques that use reconstructed meshes, i.e., all baselines in Tab. 2, we have trained a mesh decoder on the **pretrained and frozen** LiTo representation to provide an apple-to-apple comparison.  The mesh decoder is similar to that used by TRELLIS and TripoSF -- it takes as input a coarse occupancy grid ($64^3$) and LiTo representation, and decodes Flexicube coefficients on a $256^3$ grid (see Fig. S5). In contrast to both TRELLIS and TripoSF, which require a ground-truth coarse occupancy grid in the reconstruction setting, we can acquire the coarse occupancy grid directly from the LiTo representation (i.e., no ground-truth information is needed). We have added the details about the mesh decoder in Sec. E (Fig. S5).
>
> After resolving the data issues and applying the mesh decoder for fair comparison, LiTo outperforms all baselines, both with and without the ground-truth coarse geometry that TRELLIS and TripoSF rely on. Please see Fig. S1 for qualitative results.
>
> ## About the impact of depth accuracy
>
> In its current form, our encoder-decoder will produce inferior results when used with inaccurate depth maps, as it is trained with ground-truth surface points.
>
> **Note that this is not a shortcoming of our method, but rather an explicit design choice**; we aim to build a latent representation of ideal geometry and view-dependent appearance, where any ambiguities are punted to downstream models (akin to 2D latent space image representation and generation). From this perspective, the input must therefore be as high-quality and information-rich as possible, and the resultant pipeline is not meant to be applied directly to inverse problems.
>
> Nonetheless, the ability to handle more inaccurate inputs, e.g., depth and poses from a large model or real sensor, is an exciting direction that could expand both the diversity of our training data and the space of downstream applications. We hope to explore this aspect in future work.
>
> ## About fewer input views during inference
>
> Thank you for the suggestion. We evaluate LiTo and TRELLIS on a smaller number of input views, i.e., 120, 90, 60, and 30 input views uniformly sampled from the sphere. The quantitative results are presented in Tab. S5. While both TRELLIS and LiTo maintain strong performance across input view counts, LiTo results are remarkably consistent --- we believe this is because LiTo uses a fixed number of surface light-field samples, and using more views simply increases the set of points from which LiTo inputs can be selected.
>
> ## Clarification about tasks
>
> Thanks for the clarifying question. The main motivation of the paper is to learn a representation that captures complex geometries and appearances, and can be used for various downstream tasks. Single-image 3D generation and 3D reconstruction/rendering (via 3D Gaussians) are two downstream tasks where we demonstrate the effectiveness of this representation.

---

### Official Review · Reviewer_2eED · 2025-10-27

**Soundness:** 3
**Presentation:** 3
**Contribution:** 3
**Rating:** 6
**Confidence:** 4

**Summary:**

This paper proposes a 3D latent representation that jointly encodes both geometry and view-dependent color. The model takes surface locations, view directions, and colors as inputs, and encodes them into $k$ tokens of $d$ dimensions. A decoder then reconstructs geometry via predicted velocities and view-dependent Gaussian parameters. To assess the quality of the latent space, the authors train a generative model that synthesizes 3D latents from a single image. Experiments on multiple benchmarks demonstrate that the proposed representation achieves high fidelity, particularly in high-frequency details.

**Strengths:**

1. Comprehensive experiments. The experimental section is thorough, with well-designed ablation studies that clearly justify key architectural and training choices.
2. High reconstruction fidelity. The method achieves superior fidelity in input images, which is an important aspect of 3D generation quality.

**Weaknesses:**

- The view-dependent color entangles the representation with environment lighting. While this improves reconstruction fidelity, it limits the method’s applicability for tasks requiring relighting or lighting-invariant representations.

**Questions:**

1. How does the inference time compare to TRELLIS?

2. In Table 3, why does the method show worse performance on novel-view metrics (FID_dino and KID_dino)

3. Since this representation encodes view-dependent color, I’m curious how it performs on more challenging benchmarks such as Shiny Blender and NeRFactor, which involve complex lighting and reflectance conditions.

Ref-NeRF: Structured View-Dependent Appearance for Neural Radiance Fields

NeRFactor: Neural Factorization of Shape and Reflectance Under an Unknown Illumination

---

> ### Author Response · Authors · 2025-11-22
> **Responses part 1**
>
> ## About relighting
>
> We would like to emphasize that the primary goal of this work is to introduce a latent representation that faithfully encodes the geometry and view-dependent appearance of an object in its environment. Such a representation builds towards models that digitize real-world objects such that they can be experienced exactly as they originally appeared, suitable for many consumer experiences like virtual tourism and digital memory preservation. Consequently, relighting is a distinct and additional task that lies beyond the scope of this paper.
>
> Nevertheless, as we will discuss below, our work can potentially be considered as a necessary “stepping stone” towards relightable digital replicas, as inferring reflectance information from a real-world scene must fundamentally reason about the surface light field that our representation encodes.
>
> While this is in no way our primary goal, we suspect that our proposed representation may already be more useful than existing 3D latent representations for relighting tasks. We address the specific question below.
>
>
> > The view-dependent color entangles the representation with the environment lighting. While this improves reconstruction fidelity, it limits the method’s applicability for tasks requiring relighting or lighting-invariant representations.
>
> We would like to clarify that avoiding the incorporation of view-direction information does not, on its own, ensure that existing approaches can recover lighting-free albedo, nor does it necessarily simplify subsequent relighting.
>
> In fact, when learning the latent representation, since the training data contains view-dependence, existing methods have to explain view-**dependent** effects using only view-**independent** components. This mismatch makes extracting lighting-invariant albedo a nontrivial process, as lighting information becomes baked in.
>
> For instance, unnatural highlights that do not move with observer motion often appear in the outputs of such approaches.  This phenomenon is clearly visible in Fig. 4 and in reconstruction videos on the supplementary website. For example, in the first two rows of Fig. 4, the metallic top of the grinder is noticeably overexposed in TRELLIS’s results --- TRELLIS cannot explicitly represent the ground-truth view-dependent highlights (that cause the metallic top to look brighter only from certain angles) and thus can only bake these highlights into the view-independent appearance, thus incorrectly increasing the overall brightness.
>
> In other scenes in Fig. 4, slight but incorrect view-dependent effects appear. Specifically, as TRELLIS only uses 0-degree spherical harmonics, view dependence is achieved by arranging view-independent Gaussians into complex micro-surface structures (akin to microfacet BRDF theory). This mixing of incorrect appearance and complex micro-geometry again makes extracting lighting-invariant albedo difficult.
>
> In comparison, with the availability of higher degree spherical harmonics, our model can explain view-dependent effects without relying on creating microstructures using view-independent components.  To show this, we provide new qualitative examples in the appendix **Sec. F.1 (Fig. S8 and S9)**.  As we can see, when we extract and render only the 0th-degree spherical harmonics from our outputs (which model spherical harmonics up to degree 3), the results show that our 0th-degree rendering is view-independent, indicating our model utilizes higher degree spherical harmonics for view-dependent effects instead of relying on micro-surface geometry.  This indicates a cleaner separation between geometry and appearance.  Note, all the other qualitative results presented in the paper are from full 3-degree spherical harmonics.
>
> Further, we argue that modeling view-dependent appearance may actually provide more information for relighting tasks, compared to the view-independent reconstructions produced by past work.
>
> For one, proper relighting requires inferring material parameters (i.e., BRDF) for every surface location of the object. Theoretically speaking, this requires, at a minimum, reasoning about how outgoing radiance changes as a function of viewing direction, i.e., the surface light field that our representation captures. This cue is not provided by a view-independent representation.
>
> Combined with the better separation of geometry and appearance as discussed above, relighting should be an easier task with our view-dependent representation compared to previous view-independent representations.

---

> ### Author Response · Authors · 2025-11-22
> **Reponses part 2**
>
> ## About inference time
>
> Please see Tab. S8 in the updated PDF, where we provide a new runtime analysis that details the inference time for individual components used by the single-image-to-3D application.  When compared to TRELLIS, which takes 11.8 seconds to sample SLAT, it takes LiTo 9.3 seconds on an NVIDIA H100 80GB HBM3 GPU.
>
> ## About generation performance
>
> The previously reported inferior results were due to under-training: our model was evaluated at 280k iterations, whereas TRELLIS was trained for 400k iterations. We now provide the full results in the Tab. S6. As shown in rows 2-1 through 2-3 of Tab. S6, our model’s novel view synthesis performance consistently improves with additional training.
>
> We have also updated Tab. 3 accordingly: we outperform TRELLIS on most metrics, with comparable KID-DINO.
>
> ## About NeRF datasets
>
> This is an inspiring suggestion. However, after reviewing the NeRF datasets, we are unable to make a fair comparison with typical NeRF methods due to the dependence on accurate depth:
>
> - Our reconstruction, i.e., the encoder-decoder pipeline, is trained with surface light fields in Eq. (1) produced from ground-truth depth information, which NeRF datasets usually do not have.
> - Although the Ref-NeRF dataset provides disparity maps, we found that the depth quality was very low (see Fig. S10).
>
> In its current form, our encoder-decoder will produce inferior results when used with low-quality depth maps, as it is trained with ground-truth surface points. **Note that this is not a shortcoming of our method, but rather an explicit design choice**; we aim to build a latent representation of ideal geometry and view-dependent appearance, where any ambiguities are punted to downstream models (akin to 2D latent space image representation and generation). From this perspective, the input must therefore be as high-quality and information-rich as possible, and the resultant pipeline is not meant to be applied directly to inverse problems.
>
> Nonetheless, the ability to handle more inaccurate inputs, e.g., depth and poses from a large model or real sensor, is an exciting direction that could expand both the diversity of our training data and the space of downstream applications. We hope to explore this aspect in future work.

---

### Official Review · Reviewer_42pJ · 2025-10-30

**Soundness:** 3
**Presentation:** 4
**Contribution:** 3
**Rating:** 8
**Confidence:** 4

**Summary:**

The paper presents a novel autoencoder that learns a latent representation capturing the view-dependent appearance of 3D assets in addition to geometry. Besides reconstruction, the latents can be used to supervise a DiT model, and in turn can be used to generate 3D assets from a single image.

**Strengths:**

- The paper is well written and mostly easy to follow.
- Modeling of view-dependent appearance while retaining performance on geometry modeling is a novel and valuable contribution.
- Experiments and ablations are sufficient to evaluate architecture and performance. LiTo consistently achieves strong empirical results.

**Weaknesses:**

- Some architectural details are missing for reproducibility (see questions).

**Questions:**

Q1. Is the flow matching decoder only used to predict the coarse occupancy grid, which then guides the Gaussian decoder to predict the splat geometry (including xyz) and appearance? More details of the Gaussian decoder need to be provided (similar to Q3 below). Are the splat mean and covar predicted directly in world space, and are they scaled or clamped to lie within scene bounds?

Q2. What is the k used for selection of nearest neighbors for cross-attention/3D patchification?

Q3. For the generative model, what is the structure/dimensions of the learnable positional encoding? What is the patch size and architecture of the learnable patchifier? For reproducibility, authors should include a table of dimensions and structure for each module, instead of only reporting the number of parameters.

Q4. What are the inference-time memory and runtime costs for image to 3D generation?

---

> ### Author Response · Authors · 2025-11-22
>
> ## About architecture details
>
> Thanks for the suggestions. We provide details regarding the network architecture in Sec. E in the updated PDF. We provide answers to your specific questions in the following.
>
>
> > Is the flow matching decoder only used to predict the coarse occupancy grid, which then guides the Gaussian decoder to predict the splat geometry (including xyz) and appearance?
>
> Yes, this is correct
>
>
> > More details of the Gaussian decoder need to be provided (similar to Q3 below).
>
> Please refer to Fig. S4 in the updated PDF for architecture details.
>
>
> > Are the splat mean and covar predicted directly in world space,
>
> The mean (position) is predicted with respect to a normalized space centered around the occupied voxel’s world coordinates and is then translated to the world coordinate system using the voxel’s information. See Sec. E.3.
>
> The Gaussian’s rotation is predicted directly in the world coordinate system.
>
>
> > are they scaled or clamped to lie within scene bounds?
>
> No. The predicted values are not further scaled or clamped.
>
>
> > What is the $k$ used for selection of nearest neighbors for cross-attention/3D patchification?
>
> For “k samples” in L242, the number of samples equals the number of latent tokens, which is 8192 in LiTo (See Row 7 in Tab. 2).  We updated the PDF to clarify.
>
>
> > For the generative model, what is the structure/dimensions of the learnable positional encoding?
>
> The “zero-initialized learnable position encoding for each latent token” in L320 is a learnable tensor with shape of (8192, 1152), where 8192 is the number of latent tokens and 1152 is the feature dimension. The learned position encoding will be added to the projected latent tokens, i.e., feeding latent tokens through a linear layer that transforms the feature dimension from 32 to 1152. See Fig. S7.
>
>
> > What is the patch size and architecture of the learnable patchifier?
>
> The “learnable patchification layer” in L322 is a single convolution layer with the same kernel and stride size as the patch size for DINOv2-large (L321). This convolution layer will be applied to the conditioning RGB image, whose output will be concatenated with the DINOv2 features. This provides a learned-from-scratch feature that is tailored to our generation task and is complementary to the pre-trained DINOv2 feature.
>
>
> > For reproducibility, authors should include a table of dimensions and structure for each module, instead of only reporting the number of parameters.
>
> Please refer to Sec. E in the updated PDF for detailed architectures.
>
>
> ## Inference-time memory and time analysis
>
> Please refer to Tab. S8 in the updated PDF for single-image-conditioned generation memory and time analysis.  When compared to TRELLIS, which takes 11.8 seconds to sample SLAT, it takes LiTo 9.3 seconds on an NVIDIA H100 80GB HBM3 GPU.

---

### Official Review · Reviewer_2CdX · 2025-10-31

**Soundness:** 3
**Presentation:** 3
**Contribution:** 2
**Rating:** 4
**Confidence:** 3

**Summary:**

This paper propose a 3D latent representation that jointly models object geometry and view-dependent appearance from RGB-D inputs, capturing both within a unified latent space. This approach reproduces realistic effects like specular highlights and Fresnel reflections and enables high-fidelity 3D generation consistent with the input image’s lighting and materials.

**Strengths:**

1. The writing is clear and easy to follow.

2. The proposed pipeline appears novel; however, the motivation could be made more convincing or better justified.

**Weaknesses:**

Unclear motivation:
The motivation behind the proposed method remains unclear. Existing approaches typically avoid incorporating view-direction information because doing so simplifies subsequent relighting tasks. If all lighting- and view-related information are modeled jointly, it becomes questionable how the proposed model can perform relighting and be naturally integrated into a scene without introducing inconsistent illumination or lighting variations. The authors should clarify the motivation and explain how their approach maintains relighting consistency under such conditions.

Lack of ablation studies:
The current version of the paper does not include any ablation studies in the main text. A detailed ablation analysis is essential to demonstrate the contribution of each component and validate the effectiveness of the proposed method. The absence of such studies significantly weakens the paper. The authors should provide comprehensive ablation results to substantiate their design choices; without them, the paper’s claims are difficult to evaluate.

**Questions:**

please see weakness

---

> ### Author Response · Authors · 2025-11-22
> **Responses part 1**
>
> ## About motivation
>
> Our motivation is to learn “a 3D latent representation that captures both geometry and view-dependent appearances” (L035, L079), which can be utilized in “generation of full 3D objects whose appearances reflect the lighting and materials in the input” given only a single conditioning image (L085).
>
> This motivation is driven by the common pipeline in 2D image generation, e.g., StableDiffusion [a], where an image is first encoded into a latent space and a diffusion-based generative model is then trained to produce samples in that latent space. Analogously, we argue that a high-quality latent representation for the 3D domain should be able to “capture realistic view-dependent effects” (L032). However, we think “most existing 3D representations tackle only part of this problem” (L051), thus do not provide a sufficiently expressive 3D latent space. This lack of a high-quality 3D latent representation motivates our proposed method.
>
> [a] Rombach et al., High-Resolution Image Synthesis with Latent Diffusion Models. CVPR 2022.

---

> ### Author Response · Authors · 2025-11-22
> **Responses part 2**
>
> ## About relighting
>
> We would like to emphasize that the primary goal of this work is to introduce a latent representation that faithfully encodes the geometry and view-dependent appearance of an object in its environment. Such a representation builds towards models that can digitize real-world objects such that they can be experienced exactly as they originally appeared, suitable for many consumer experiences like virtual tourism and digital memory preservation. Consequently, relighting is a distinct and additional task that lies beyond the scope of this paper.
>
> Nevertheless, as we will discuss below, our work can potentially be considered as a necessary “stepping stone” towards relightable digital replicas, as inferring reflectance information from a real-world scene must fundamentally reason about the surface light field that our representation encodes.
>
> While this is in no way our primary goal, we suspect that our proposed representation may already be more useful than existing 3D latent representations for relighting tasks. In particular, we respectfully disagree with the reviewer on the following statement:
>
>
> > Existing approaches typically avoid incorporating view-direction information because doing so simplifies subsequent relighting tasks.
>
> We would like to clarify that avoiding the incorporation of view-direction information does not, on its own, ensure that existing approaches can recover lighting-free albedo, nor does it necessarily simplify subsequent relighting.
>
> In fact, when learning the latent representation, since the training data contains view-dependence, existing methods have to explain view-**dependent** effects using only view-**independent** components. This mismatch makes extracting lighting-invariant albedo a nontrivial process, as lighting information becomes baked in.
>
> For instance, unnatural highlights that do not move with observer motion often appear in the outputs of such approaches.  This phenomenon is clearly visible in Fig. 4 and in reconstruction videos on the supplementary website. For example, in the first two rows of Fig. 4, the metallic top of the grinder is noticeably overexposed in TRELLIS’s results --- TRELLIS cannot explicitly represent the ground-truth view-dependent highlights (that cause the metallic top to look brighter only from certain angles) and thus can only bake these highlights into the view-independent appearance, thus incorrectly increasing the overall brightness.
>
> In other scenes in Fig. 4, slight but incorrect view-dependent effects appear. Specifically, as TRELLIS only uses 0-degree spherical harmonics, view dependence is achieved by arranging view-independent Gaussians into complex micro-surface structures (akin to microfacet BRDF theory). This mixing of incorrect appearance and complex micro-geometry again makes extracting lighting-invariant albedo difficult.
>
> In comparison, with the availability of higher degree spherical harmonics, our model can explain view-dependent effects without relying on creating microstructures using view-independent components.  To show this, we provide new qualitative examples in the appendix **Sec. F.1 (Fig. S8 and S9)**.  As we can see, when we extract and render only the 0th-degree spherical harmonics from our outputs (which model spherical harmonics up to degree 3), the results show that our 0th-degree rendering is view-independent, indicating our model utilizes higher degree spherical harmonics for view-dependent effects instead of relying on micro-surface geometry.  This indicates a cleaner separation between geometry and appearance.  Note, all the other qualitative results presented in the paper are from full 3-degree spherical harmonics.
>
> > If all lighting- and view-related information are modeled jointly, it becomes questionable how the proposed model can perform relighting and be naturally integrated into a scene without introducing inconsistent illumination or lighting variations.
>
> We would like to provide an alternative argument that modeling view-dependent appearance may actually provide more information for relighting tasks, compared to the view-independent reconstructions produced by past work.
>
> For one, proper relighting requires inferring material parameters (e.g., BRDF) for every surface location of the object. Theoretically speaking, this requires, at a minimum, reasoning about how outgoing radiance changes as a function of viewing direction, i.e., the surface light field that our representation captures. This cue is not provided by a view-independent representation.
>
> Combined with the better separation of geometry and appearance as discussed above, relighting should be an easier task with our view-dependent representation compared to previous view-independent representations.

---

> ### Author Response · Authors · 2025-11-22
> **Responses part 3**
>
> ## About ablations
>
> Thanks for the suggestion, we will bring essential ablations to the main paper and provide clear references to ablations in the appendix.
>
> In the meantime, we want to mention that we have detailed ablations in the appendix as stated in L424. In our originally-submitted version, we include the following ablations:
>
> - reconstruction results on 3 different datasets under 3 different lightings, i.e., 9 evaluation sets with millions of evaluation images: Tab. S2, S3, and S4 study the effects of different lighting, and they show our representation is robust to light conditions.
> - the effect of using higher order degrees of spherical harmonics in reconstruction training, e..g, Row 3-3 - 3-6 in Tab. S4: higher-order degree enhances appearance modeling qualities;
> - the effect of known (and unknown) conditioning camera parameters during generative model training, i.e., Row 2-1 vs 3 in Tab. S6: ray information is not needed.
>
> As recommended, we further enhance the ablation studies with new studies in the updated PDF:
>
> - the effects of including view information as encoder’s input, e.g., Row 3-6 vs 3-7 in Tab. S4. It studies the effect of modeling a point cloud or a surface light field. As shown, modeling surface light field improves appearance and geometry modeling for objects under complex lighting.
> - whether to use the flow-matching velocity decoder to get geometry or a separately-trained occupancy decoder (L311), e.g., Row 3-6 vs 3-8 in Tab. S4: separately-trained occupancy prediction does not harm the performance but improves the speed;
> - whether to use a mesh decoder, e.g., Row 3-8 vs 3-9 in Tab. S4: mesh decoder improves geometry quality.
> - whether changing the number of input views affects reconstruction quality, e.g., Tab. S5: our approach is robust to the number of input views.
> - whether our generative model is robust to ODE integration algorithms and settings during inference, i.e., Tab. S7: we observe consistent performance across algorithms and settings.

---

### Author Response · Authors · 2025-11-22

We thank all reviewers’ time and effort in reviewing our paper.

We are encouraged that reviewers acknowledge that our paper is “easy to follow” (**2CdX, 42pJ**) and enables “easy comprehension” (**wBM6**).

We are also happy that the reviewers think the proposed approach is “novel” (**2CdX, 42pJ**), is “the first work to unify 3D geometry and view-dependent appearance in a single framework” (**wBM6**), and that the reviewers appreciate the “superior fidelity” in reconstruction (**2eED**) and “consistently achieves strong empirical results” (**42pJ**).

We are delighted to see that reviewers find “experiments and ablations are sufficient” (**42pJ**), "comprehensive experiments" (**2eED**), and "the experimental design is rigorous" (**wBM6**).

We address reviewers’ questions below.

---

### Meta-Review · Area_Chair_jxW5 · 2026-01-06

**Summary:**

The paper proposes LiTo, a method for 3D latent representation that jointly models object geometry and view-dependent appearance. Unlike previous works often focus on diffuse color, LiTo encodes surface light field into compact latent vectors. They further train a latent flow matching model conditioned on the image input to generate 3D objects consistent with the lighting and materials.

**Strength**
- Novelty: Reviewers agree on the novelty of this work, since it is the first to unify 3D geometry and view-dependent appearance in a single framework via surface light field tokenization.
- High Fidelity: The method can capture high-frequency details and view-dependent effects
- Strong results: the method achieves SoTA performance in geometry and generation tasks, outperforming baselines like TRELLIS.

**Weakness and Rebuttal Resolution**:
- Relighting concerns: reviewers asked whether encoding view-dependent appearance would harm relighting performance. The authors argued that the use of higher-order SH allows for better separation of geometry and appearance compared to methods that "bake in" lighting artifacts.
- Fair comparison: This has been addressed by clarifying differences in input data, since baselines use GT coarse geometry. The authors implemented a mesh decoder for a fair comparison and showed that their method outperforms baselines
- More ablation: authors provided extensive additional ablation studies.

The comprehensive rebuttal leads to a consensus among all reviewers (scores of 8, 6, 6, and reviewer 2CdX raising the score from 4 to 6), which also indicates that the work is solid. Therefore, I recommend accepting the paper.

**Reviewer Concerns:**

I believe the rebuttal has resolved most of concerns.

**Reviewer Scores:**

All reviewers will probably keep their positive, except for the only negative reviewer 2CdX, where they explicitly mentioned that they are inclined to raise the score to 6.

---

### Decision · Program_Chairs · 2026-01-26

Accept (Poster)